# Scheduling Your LLM Reinforcement Learning with Reasoning Trees

**Hong Wang**[1*] **Zhezheng Hao**[2*] **Jian Luo**[3] **Chenxing Wei**[1]

**Yao Shu**[4] **Lei Liu**[3] **Qiang Lin**[1] **Hande Dong**[1†] **Jiawei Chen**[2†]

[1] Tencent    [2] Zhejiang University    [3] Independent Researcher
[4] Hong Kong University of Science and Technology (Guangzhou)
{hwhongwang, handedong}@tencent.com
{haozhezheng, sleepyhunt}@zju.edu.cn

## Abstract

Using Reinforcement Learning with Verifiable Rewards (RLVR) to optimize Large Language Models (LLMs) can be conceptualized as progressively editing a query's 'Reasoning Tree'. This process involves exploring nodes (tokens) and dynamically modifying the model's policy at each node. When combined with data scheduling, this process yields further gains in data efficiency and accuracy. However, existing RLVR data scheduling methods typically rely on path-based metrics to rank queries, overlooking the reasoning tree structures of these queries. In this paper, we introduce a novel metric, namely **Reasoning Score** (r-score), which measures the query's learning difficulty based on the structure of its reasoning tree. Based on the r-score, we propose the **Reasoning Tree Schedule** (Re-Schedule), a scheduling algorithm that constructs a curriculum progressing from structurally simple (high r-score) to complex (low r-score) queries. Experiments on six math-reasoning benchmarks show that Re-Schedule significantly improves average accuracy, achieving gains of up to 3.2%. These strong results validate our approach and demonstrate that a structural understanding of the reasoning tree provides a more powerful and principled foundation for RLVR data scheduling[1].

## 1 Introduction

Advancing the complex reasoning capabilities of Large Language Models (LLMs) remains a significant challenge, particularly in domains like mathematical problem-solving. Reinforcement Learning with Verifiable Reward (RLVR) (Gao et al., 2024; DeepSeek-AI et al., 2025), especially through policy optimization methods like GRPO (Shao et al., 2024), has emerged as a powerful paradigm to address this challenge. As shown in Figure 1 (a), in this framework, the space of potential solution paths for a query can be modeled as a specific 'Reasoning Tree' (Wang et al., 2025e; Yang et al., 2025b), where each node represents an intermediate reasoning step and each path represents a potential solution trajectory. From this perspective, RLVR operates as a dynamic 'node-editing' process of the reasoning tree: by rewarding correct paths and penalizing incorrect ones, the model iteratively refines its decision policy at each tree node. This optimization process gradually prunes branches that lead to low-quality or incorrect solutions, thereby improving overall reasoning accuracy.

In this paradigm, data scheduling plays a critical role in model performance (Hu et al., 2025; Li et al., 2025; Yang et al., 2026; Ren et al.). The concept of data scheduling originates from curriculum learning (Bengio et al., 2009), which posits that models learn more effectively when training examples (queries) are organized in a meaningful sequence. Existing data scheduling strategies typically pre-define a 'difficulty' metric for queries, and and schedule them from easy to hard to improve data efficiency and final performance (Xi et al., 2024; Chen et al., 2025b;a; Dai et al., 2025) However, from a reasoning tree perspective, current difficulty measure strategies exhibits a critical limitation: current methods estimate difficulty primarily via final solution accuracy, overlooking richer

---

*Equal Contribution.
†Corresponding Authors
[1]Our code is available at https://github.com/zz-haooo/Re-Schedule.

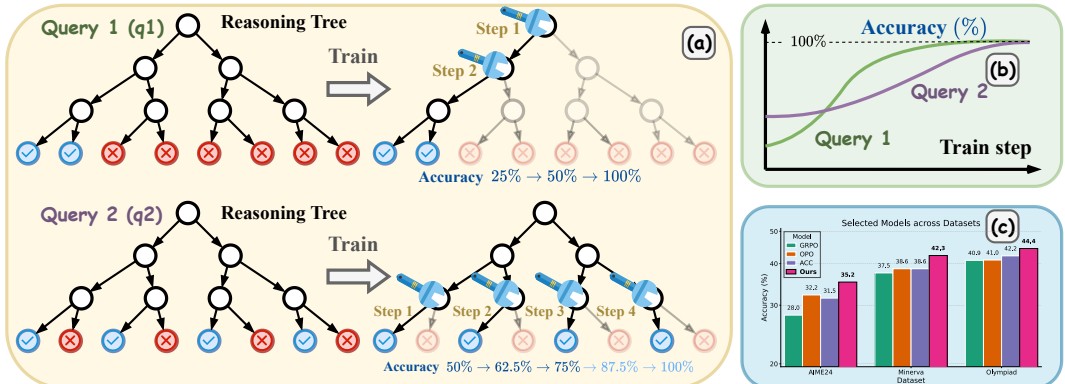

Figure 1: **(a)** A simple reasoning tree (q1) requires less node editing for performance improvement than a complex one (q2). **(b)** Consequently, q1 shows high training efficiency (steep learning curve) despite low initial accuracy, while q2's complex structure leads to low efficiency. **(c)** Our method leverages this structural insight to significantly outperform baselines on various datasets.

query-level characteristics such as the structural complexity of the reasoning tree. Accuracy alone is insufficient — low accuracy does not necessarily indicate that a query is inherently hard, and high accuracy does not guarantee ease of optimization. This inconsistency can undermine the efficacy of accuracy-based scheduling approaches. We illustrate this issue with the following examples.

To illustrate, consider two representative queries, **q1** and **q2**, whose reasoning trees are shown in Figure 1(a). As depicted in Figure 1(b), LLMs may exhibit low initial accuracy on **q1**, due to the presence of many incorrect solution trajectories (reasoning paths). However, its simple tree structure means that modifying a few key decision nodes can yield substantial accuracy gains, indicating high learning efficiency despite the poor initial performance. In contrast, **q2** achieves higher initial accuracy, with roughly half of its trajectories being correct, yet these correct paths are scattered across disparate subtrees. This fragmented structure requires more extensive edits across numerous tree nodes, typically resulting in higher training difficulty and lower learning efficiency. Critically, existing path-based metrics will misinterpret **q1**'s low accuracy as high difficulty, thus assigning it a lower training weight, while incorrectly prioritizing the more difficult **q2**. Such path-based metrics may lead to a less efficient training process. This motivates our central research question: How can we move beyond path-based metrics to directly quantify a query's true learning difficulty from its reasoning-tree structure?

To address this question, we introduce the **Reasoning Score** (r-score), a novel metric that quantifies a query's learning potential based on its reasoning tree structure. We formalize this by framing the reinforcement learning training process as an optimization problem under a finite 'node editing budget', which we define as a fixed number of node editing operations. **Consequently, a query's r-score is its maximum potential accuracy gain achievable within this limited editing budget.** This metric clearly explains the discrepancy in our example: **q1**, with its 'concentrated' error structure, yields a high r-score because a small budget (e.g., two edits) produces a massive accuracy gain (+75%). Conversely, **q2**'s 'diffuse' structure results in a low r-score, as the same budget only yields marginal improvement (+25%). Therefore, a higher r-score signifies a more tractable reasoning structure and greater learning efficiency, offering a more comprehensive assessment of difficulty than path-based metrics.

Building on the Reasoning Score, we propose the **Reasoning Tree Schedule (Re-Schedule)**, a novel data scheduling algorithm designed to guide RLVR training more efficiently. Our method consists of three main stages. First, an offline approximation of each query's reasoning tree is constructed by sampling multiple solution trajectories from a base model. Second, this approximated reasoning tree is used to calculate each query's reasoning score by simulating the editing process. Finally, we integrate the r-score as a dynamic weight into the RLVR loss function to form a schedule. This schedule prioritizes high-scoring (simple) queries in the initial training phases to accelerate convergence on simple queries. As training progresses, the weighting gradually shifts to lower-scoring (difficult) queries, enabling the model to master more challenging problems.

In summary, the main contributions of this paper are:

- We introduce the Reasoning Score (r-score), a new tree-based metric that measures a query's learning efficiency rather than its path-based solution accuracy.
- We propose Re-Schedule, a data scheduling algorithm that uses the r-score to create an effective, easy-to-hard curriculum for RLVR.
- As shown in Figure 1(c), we empirically demonstrate that our approach significantly improves average accuracy, achieving gains of up to 3.2%, on complex reasoning tasks.

## 2 RELATED WORK

### 2.1 REINFORCEMENT LEARNING WITH VERIFIABLE REWARDS IN LLMS

Reinforcement learning with verifiable reward (RLVR), where the reward is computed by a rule-based verification function, has been shown to be effective in improving the reasoning capabilities of LLMs (Gao et al., 2024; DeepSeek-AI et al., 2025; Kimi et al., 2025; Zeng et al., 2025; Wen et al., 2025; Song et al., 2025; Wei et al., 2025b;a;e;d; Ma et al.). Typically, RLVR frameworks assign a binary reward by comparing the model's generated output against a ground-truth solution, indicating whether it is correct or incorrect. This reward design obviates the need for complex outcome-based or process-based reward models, offering a straightforward yet potent approach. Recent advancements in policy optimization algorithms, such as PPO and GRPO, have further refined this paradigm (Schulman et al., 2017; Kazemnejad et al., 2024; Yuan et al., 2025; Yue et al., 2025; Shao et al., 2024; Yu et al., 2025; Liu et al., 2025; Zhang et al., 2025a; Hu, 2025; Hao et al., 2025b; Wei et al., 2025c; Liu et al., 2026; Yang et al., 2025a). In contrast to these studies, which focus on algorithmic improvements, our work builds upon the standard GRPO framework with a primary focus on designing a more effective training data schedule.

### 2.2 DATA SCHEDULING ALGORITHM IN LLM REINFORCEMENT LEARNING

Various data scheduling strategies have been proposed to enhance the reasoning capabilities in LLM Reinforcement Learning. These can be broadly categorized into static selection and dynamic adjustment methods. Representative of static selection is LIMR (Li et al., 2025), which selected 1.4k examples from an 8.5k set for RLVR to match the performance of using the full set. In contrast, dynamic strategies make real-time adjustments during training. For instance, $R^3$ employs reverse curriculum reinforcement learning to simplify the model's exploration space (Xi et al., 2024). LPPO (Chen et al., 2025b) utilize the gradient of accuracy to prioritize data, effectively treating learning difficulty as a derivative of performance. Similarly, Seed-GRPO (Chen et al., 2025a) employs semantic diversity (uncertainty) as a proxy for difficulty. Furthermore, DELT leverages training gradients to measure the quality and learnability of data (Dai et al., 2025), subsequently adjusting sample weights. However, these methods rely on outcome-based proxies (e.g., accuracy), effectively treating reasoning as a flat sequence (Zhao et al., 2025; Zhang et al., 2025b). They overlook the inherent tree-structured solution space of reasoning tasks. In contrast, our approach explicitly leverages this topological structure. By analyzing the Reasoning Tree, we directly quantify a query's 'structural learnability', providing a more precise and principled measure of difficulty than performance statistics alone.

## 3 PRELIMINARIES

### 3.1 GROUP RELATIVE POLICY OPTIMIZATION

The objective of the GRPO algorithm is to optimize a policy $\pi_\theta$ based on a group of generated responses (Shao et al., 2024; Yu et al., 2025). For a query $q$ from a dataset $\mathcal{D}$, the policy generates $G$ responses $\{o_i\}_{i=1}^G$. The token-level objective function is formulated as:

$$\mathcal{J}(\theta) = \mathbb{E}_{q \sim \mathcal{D}, \{o_i\}_{i=1}^G \sim \pi_{\text{old}}(\cdot|q)} \left[ \frac{1}{\sum_{i=1}^G |o_i|} \sum_{i=1}^G \sum_{t=1}^{|o_i|} \min\left(r_{i,t} A_{i,t}, \text{clip}(r_{i,t}, 1-\varepsilon, 1+\varepsilon) A_{i,t}\right) \right],$$

(1)

where $r_{i,t} = \frac{\pi_\theta(o_{i,t}|q,o_{i,<t})}{\pi_{\text{old}}(o_{i,t}|q,o_{i,<t})}$ is the probability ratio of the token $o_{i,t}$ between the current and old policies. The advantage term $A_{i,t}$ is constant for all tokens within a single response and is calculated by normalizing the response's reward $R_i$ relative to the other responses in the group:

$$A_{i,t} = \frac{R_i - \text{mean}(\{R_k\}_{k=1}^G)}{\text{std}(\{R_k\}_{k=1}^G) + \delta}, \quad \forall t, \tag{2}$$

where $\delta$ is a small constant for numerical stability.

Data scheduling algorithms can be formulated by introducing a weighting function $\omega(q,t)$ that modulates the contribution of each query $q \in \mathcal{D}$ and current epoch $t$ to the overall objective. Specifically, the objective in Equation 1 is modified as follows:

$$\mathcal{J}_{\text{schedule}}(\theta) = \mathbb{E}_{q\sim\mathcal{D},\dots} [\omega(q,t) \cdot (\text{original objective term for } q)]. \tag{3}$$

Note: In the equations above, we have abbreviated the full objective for clarity. For example, in an accuracy-based curriculum learning, the training weight $\omega$ is formulated as a function of the query's accuracy $\text{ACC}(q)$ and current epoch $t$:

$$\alpha(\text{ACC}(q), t) = (1 - \gamma(t))\text{ACC}(q) + \gamma(t)(1 - \text{ACC}(q)), \tag{4}$$

$$\omega = \text{rank}(\alpha)\% \cdot \omega_{\max} + (1 - \text{rank}(\alpha)\%) \cdot \omega_{\min}. \tag{5}$$

Here, $\omega_{\max}$ and $\omega_{\min}$ are hyperparameters that define the maximum and minimum training weights (e.g., $\omega_{\max} = 0.8$, $\omega_{\min} = 0.2$); And $\text{rank}(\alpha)$ means calculating the reverse order of $\alpha$ in the entire dataset. The term $\gamma(t)$ is a scheduling function that progresses over time. Common choices for $\gamma(t)$ include a linear mapping, $\gamma(t) = t/T$, or a sigmoid function, $\gamma(t) = \sigma\left(\left(\frac{t}{T} - 0.5\right)\right)$, $\sigma(x) = (1 + e^{-x})^{-1}$, where $T$ is the total number of epochs.

## 3.2 REASONING TREE

For complex reasoning tasks, the process of generating a solution can be conceptualized as traversing a 'Reasoning Tree'. In this context, the root of the tree is the initial prompt, and each node represents a partial solution or an intermediate reasoning step. The branches extending from a node correspond to the possible next tokens or thought segments that the LLM can generate.

Due to the combinatorial explosion of possible solution paths, the complete reasoning tree is typically computationally intractable. Therefore, analysis often relies on a structured approximation (e.g., a fixed-structure k-ary reasoning tree). Formally, an approximated reasoning tree is defined as a triplet $T = (\mathcal{N}, \mathcal{E}, \mathcal{R})$, where $\mathcal{N}$ is the set of nodes, $\mathcal{E}$ is the set of edges, and $\mathcal{R}$ defines the parent-child relationships.

The components of the tree are described using the following notation: $\mathcal{N}_{\text{leaf}} \subset \mathcal{N}$ is the set of leaf nodes; For a given node $n_i \in \mathcal{N}$, $C(n_i)$ denotes the set of its immediate children and $\mathcal{L}(n_i)$ denotes the set of its leaf descendants. If $n_i$ is a leaf node, then $\mathcal{L}(n_i) = \{n_i\}$. Within this framework, each non-leaf node $n_i \in \mathcal{N} \setminus \mathcal{N}_{\text{leaf}}$ represents a partial reasoning path, while a complete path to a leaf node $n_j \in \mathcal{N}_{\text{leaf}}$ corresponds to a full solution trajectory.

From this perspective, the RLVR optimization process is a dynamic 'node editing' of this reasoning tree. By rewarding correct paths and penalizing incorrect ones, the policy optimization algorithm adjusts the token probabilities at each node, effectively strengthening the branches that lead to correct answers and weakening those that lead to errors. The structure of this tree—the distribution of correct and incorrect paths—is intrinsic to each problem sample and, as we will argue, is a key clue to its learning dynamics.

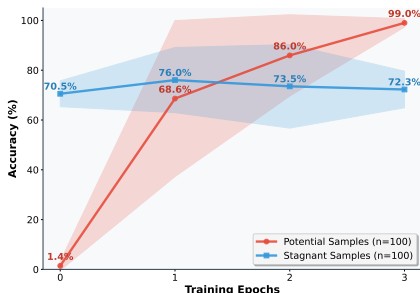

Figure 2: Accuracy Progression During Training. The solid line represents the average accuracy, and the shaded area indicates the range.

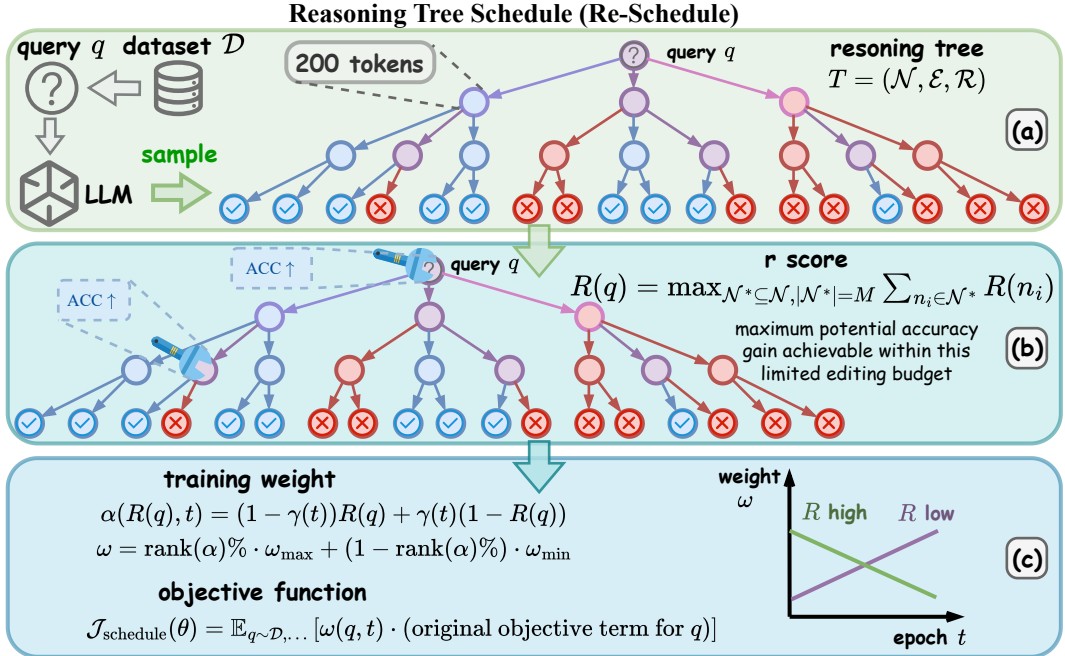

Figure 3: Overview of the **Reasoning Tree Schedule (Re-Schedule)** Algorithm.**(a) Tree Construction**: For each query, an approximate reasoning tree is constructed by sampling multiple solution paths from a base model (Note: This figureis for illustrative purposes only; our experiments use a tree with a depth of 4 and a width of 4, i.e., $k = 4, d = 4$.). **(b) R-Score Calculation**: The tree's structure is analyzed to compute the r-score, a metric quantifying the query's learning potential. **(c) Dynamic Weighting**: The r-scores are used to dynamically weight each query during training, forming a curriculum that progresses from structurally simple (easy) to complex (hard) examples.

## 4 MOTIVATION

The premise of this work is that path-based metrics such as accuracy are poor indicators of a query's true learning difficulty. To illustrate our point, we supplement the example from the introduction with an experiment. As shown in Figure 2, we selected two distinct sets of 100 queries each from the DAPA-Math-17K dataset, using the Qwen2.5-Math-7B model. The blue line represents 'Stagnant Samples'—queries with high initial accuracy but complex reasoning structures (low r-score). Their flat learning curve indicates that despite high initial performance, they are difficult to improve further. In contrast, the red line represents 'Potential Samples'—queries with low initial accuracy but simple tree structures (high r-score). Their steep learning curve demonstrates high learnability, where a small amount of training yields significant gains. This discrepancy highlights that path-based metrics, like accuracy, are biased measurements for learning difficulty. This finding motivates us to design a new metric based on the structure of the reasoning tree.

## 5 METHOD

As illustrated in Figure 3, the **Reasoning Tree Schedule (Re-Schedule)** enhances reinforcement learning performance by creating a curriculum based on our novel metric, the **Reasoning Score (r-score)**. The r-score quantifies a query's learning difficulty a priori based on the structure of its reasoning tree. Next, we will introduce the specific implementation details.

### 5.1 TREE CONSTRUCTION

As the entire reasoning tree is computationally intractable, we construct a manageable, fixed-structure $k$-ary approximation for each query $q$. The structure of this tree, $\mathcal{T}$, is defined by a branching factor $k$, a maximum depth $d$, and a token interval $l$ (e.g., $k = 4, d = 4, l = 200$).

The construction process begins at the root node (the query $q$) and proceeds via a periodic branching strategy during response generation. Specifically, a branch is triggered immediately at the beginning of the response and subsequently at every $l$-token interval. As shown in Figure 3 (a), at each trigger, the current path splits into $k$ independent sub-paths that continue to generate in parallel. This recursive branching process continues until a predefined maximum depth $d$ is reached. To minimize computational overhead from this multi-path sampling, we use the Key-Value (KV) Cache, as all sibling branches share the same prefix.

In RLVR tasks, a solution's quality is determined by the correctness of its final answer, which corresponds to a leaf node in our framework. Therefore, we define the quality of any intermediate node $n_i$ as the average accuracy of its leaf descendants, $\mathcal{L}(n_i)$. This is quantified using an accuracy function:

$$\text{ACC}(S) = \frac{\sum_{n_j \in S} \mathbb{I}(n_j \text{ is correct})}{|S|}, \tag{6}$$

where $S$ is a set of leaf nodes and $\mathbb{I}(\cdot)$ is the indicator function. This allows us to assess quality at different levels: the quality of a reasoning segment via $\text{ACC}(\mathcal{L}(n_i))$ and the model's aggregate performance on the query via $\text{ACC}(\mathcal{N}_{\text{leaf}})$.

## 5.2 R-Score Calculation

The r-score quantifies the learning potential of a node or query by measuring the maximum achievable accuracy gain under a limited policy refining cost, like a limited node editing budget. Given this idea, for any non-leaf node $n_i$, we define its r-score, $R(n_i)$, as the maximal accuracy gain achievable by selecting its single best child branch and pruning all others. This is formulated as:

$$R(n_i) = \max_{n_{\text{child}} \in \mathcal{C}(n_i)} \text{ACC}\big[\mathcal{N}_{\text{leaf}} \setminus \mathcal{L}(n_i) \cup \mathcal{L}(n_{\text{child}})\big] - \text{ACC}\big[\mathcal{N}_{\text{leaf}}\big]. \tag{7}$$

The overall r-score for a query, $R(q)$, estimates the total accuracy gain achievable under a budget that limits modifications to a maximum of $M$ nodes. It is the maximum sum of r-scores from any set of $M$ non-conflicting nodes (e.g., for a budget of $M = 4$):

$$R(q) = \max_{\mathcal{N}^* \subseteq \mathcal{N}, |\mathcal{N}^*| = M} \sum_{n_i \in \mathcal{N}^*} R(n_i). \tag{8}$$

Two nodes are considered conflicting if one is located in a subtree that is implicitly pruned by the optimal branch selection of the other.

Intuitively, solving Equation (7) represents the evaluation process of the sub-tree's structure, while a simpler structure of reasoning tree starting from $n_i$ yields a higher $R(n_i)$. Combining the evaluation $R(n_i)$ of each node $n_i$ under a limited budget $M$, solving Equation (8) is to find the maximum achievable accuracy gain over the reasoning tree, like exploring possible combinations of $M$ nodes and picking the best combination. Thus, a higher $R(q)$ indicates that substantial accuracy improvements can be made by correcting just a few critical reasoning steps, signifying a structurally simple and efficient-to-learn query.

## 5.3 Dynamic Weighting

To strike a balance between data diversity and data scheduling, we propose a weighted scheduling framework that dynamically adjusts data prioritization. Specifically, queries are assigned adaptive weights determined by both training step $t$ and r-score $R$. Specifically, when it is an early training stage, higher weights are assigned to samples with higher r-scores (indicating lower learning difficulty), stabilizing the reinforcement learning. When RL training meets the later training phase, queries' weights will be redistributed gradually towards lower-r-score samples (higher learning difficulty) to enhance model generalization.

Motivated by this, the training weight $\omega$ of each query is formulated as

$$\alpha(R(q), t) = (1 - \gamma(t))R(q) + \gamma(t)(1 - R(q)), \tag{9}$$

$$\omega = \text{rank}(\alpha)\% \cdot \omega_{\max} + (1 - \text{rank}(\alpha)\%) \cdot \omega_{\min}, \tag{10}$$

where $t$ is the current epoch; $\omega_{\max}$ and $\omega_{\min}$ are hyperparameters that define the maximum and minimum training weights; And $\text{rank}(\alpha)$ means calculating the reverse order of $\alpha$ in the entire dataset;

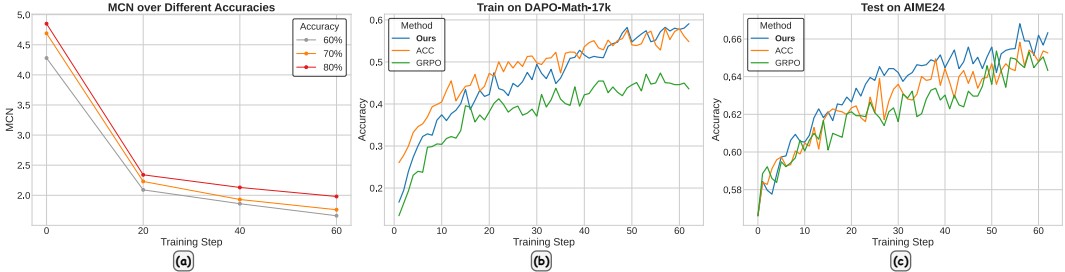

Figure 4: **(a)** The average MCN decreases over time, indicating successful tree optimization. **(b)** & **(c)** To compare metrics, we train models on the top 1/3 of data selected by each. The plots show the resulting **(b)** training accuracy and **(c)** test accuracy. The model used is Qwen2.5-Math-7B.

$\gamma(t)$ can be either linear mapping $\gamma(t) = \frac{t}{T}$ or sigmoid $\gamma(t) = \sigma\left(\left(\frac{t}{T} - 0.5\right)\right)$. The $\alpha(R(q), t)$ is a monotonically varying function that down-weights high-scoring (simple) queries over time while up-weighting lower-scoring (difficult) ones. This scheduling approach balances exploitation of easily learnable patterns and exploration of challenging instances, mitigating catastrophic forgetting of underrepresented data distributions.

# 6 ANALYSIS

## 6.1 TRAINING AS REASONING TREE OPTIMIZATION

To empirically validate that the training process is optimizing reasoning trees, we conducted an experiment centered on a new metric: the Minimum Corrective Nodes (MCN). This metric is defined as the minimum number of node modifications required for the reasoning tree to achieve a specified target accuracy. A single node modification is counted as one token change; thus, a lower MCN signifies a well-structured reasoning tree. In our experiment, we tracked the MCN on the DAPA-Math-17K training set during the training of Qwen2.5-Math-7B, excluding queries where the base model's accuracy was below 10%.

As shown in Figure 4(a), the average MCN across the training set exhibits a consistent downward trend as training progresses, regardless of the target accuracy. This result demonstrates that the reinforcement learning process effectively refines the model's policy at critical decision nodes, thereby validating our central assumption that training is a process of reasoning tree optimization.

## 6.2 THE RELATIONSHIP BETWEEN R-SCORE AND LEARNING DIFFICULTY

In this experiment, we want to see which metric best identifies valuable queries for early-stage training. The process is as follows: First, we use each metric to select the top one-third of the data, creating several distinct subsets. Second, we train a separate model on each of these subsets for a single epoch. Finally, we evaluate the resulting models on both the training and test sets.

As shown in Figure 4(b), the subset selected by the ACC-based method initially shows higher average accuracy on the training set, as expected from its selection criteria. However, as training progresses, the model trained on the r-score-selected subset quickly surpasses it. This indicates that the r-score is more effective at identifying queries with low learning difficulty, rather than just initial accuracy.

The advantage of r-score is even more evident on the test set, as shown in Figure 4(c). Here, the model trained on the r-score-selected queries consistently outperforms both the ACC-based selection and a baseline with random query selection (GRPO). This confirms that the queries identified by the r-score provide the most effective learning signal, leading to better performance improvement and validating its capability in identifying the real difficulty of queries.

# 7 EXPERIMENT

## 7.1 RL TRAINING SETUPS

**Training setting** We conduct experiments on two different models, including Qwen2.5-Math-7B and Qwen2.5-7B. We adapt our training codebase from verl (Sheng et al., 2025) and follow the training recipe of standard GRPO. Our training data is DAPO-Math-17k (Yu et al., 2025), containing only math problems with integer ground-truth answers. Both the KL-divergence and entropy loss terms are removed in our experiments (Hao et al., 2025b). Generation batch size is set to 512. Training is performed with top-p value of 1.0 and temperature = 1.0.

**Evulation** We evaluated our models and baselines on six widely used mathematical reasoning benchmarks: AIME24, AIME25, AMC23 (Li et al., 2024), MATH-500 (Hendrycks et al., 2021), Minerva Math (Lewkowycz et al., 2022), and OlympiadBench (He et al., 2024). Validation is performed with a top-p value of 0.7 and temperature = 1.0 across all models and test sets. We use Math-Verify for training, validation, and final evaluation. We report avg@32 for all datasets. All results are presented as percentages.

**Baselines** For the throughout comparison, we compare our method against 7 baselines, including standard GRPO (Shao et al., 2024), SimpleRL-Zoo (Zeng et al., 2025), Eurus-PRIME(Cui et al., 2025), OPO (Hao et al., 2025a), ACC (curriculum learning based on accuracy, using sigmoid weighting), LPPO (Chen et al., 2025b), and Seed-GRPO (Chen et al., 2025a).

**Our Methods** Re-Schedule is implemented with two weighting schemes: 'linear' and 'sigmoid'. Unless otherwise specified, the reasoning trees in our experiments are constructed with a branching factor of $k = 4$, a maximum depth of $d = 4$, and a token interval of $l = 200$. The weighting schemes are defined as follows: 1. The 'linear' scheme uses $\gamma(t) = t/T$; 2. The 'sigmoid' scheme uses $\gamma(t) = \sigma\left(\left(\frac{t}{T} - 0.5\right)\right)$. For both, we set the total number of epochs $T = 10$. Details of the training setup can be found in the Appendix C.

## 7.2 MAIN EXPERIMENT

Table 1: Main benchmark results on **Qwen2.5-Math-7B**. All values are accuracies multiplied by 100. Best results are in **bold**.

| Model | AIME24 | AIME25 | AMC23 | MATH500 | Minerva | Olympiad | Avg. |
|---|---|---|---|---|---|---|---|
| Qwen2.5-Math-7B | 13.8 | 5.3 | 44.6 | 39.6 | 9.9 | 13.8 | 21.2 |
| **Classical RLVR Methods** | | | | | | | |
| GRPO | 28.0 | 14.3 | 66.2 | 78.6 | 37.5 | 40.9 | 44.3 |
| SimpleRL-Zoo | 30.8 | 14.2 | 65.4 | 79.2 | 37.1 | 40.8 | 44.6 |
| Eurus-PRIME | 20.9 | 13.0 | 65.2 | 79.8 | 37.5 | 40.6 | 42.8 |
| OPO | 32.2 | 13.4 | 71.5 | **82.2** | 38.6 | 41.0 | 46.5 |
| **Scheduling Methods** | | | | | | | |
| ACC$_{sigmoid}$ | 31.5 | 15.6 | 70.9 | 80.8 | 38.6 | 42.2 | 46.6 |
| LPPO | 32.8 | 14.9 | 63.3 | 79.2 | 39.0 | 40.6 | 45.0 |
| Seed-GRPO | 30.7 | 14.0 | 71.0 | 80.0 | 38.2 | 38.5 | 45.4 |
| **Our Methods** | | | | | | | |
| Re-Schedule$_{linear}$ | 34.2 | 15.6 | **72.4** | 81.2 | 36.4 | 42.5 | 47.1 |
| Re-Schedule$_{sigmoid}$ | **35.2** | **16.0** | 72.3 | **82.2** | **42.3** | **44.4** | **48.5** |

As shown in Tables 1 and 2, our Re-Schedule method consistently sets a new state-of-the-art, achieving average scores of 48.5 on Qwen2.5-Math-7B and 44.5 on Qwen2.5-7B. It significantly outperforms both scheduling baselines like ACC$_{sigmoid}$ (by up to 3.2%) and classical RLVR methods like OPO/GRPO (by up to 3.8%). These results validate our central claim: that the reasoning tree's structure, captured by our r-score, is a more effective way to measure the real learning difficulty of a query than path-based metrics like accuracy.

Table 2: Main benchmark results on **Qwen2.5-7B**. All values are accuracies multiplied by 100. Best results are in **bold**.

| Model | AIME24 | AIME25 | AMC23 | MATH500 | Minerva | Olympiad | Avg. |
|---|---|---|---|---|---|---|---|
| Qwen2.5-7B | 5.1 | 2.5 | 27.8 | 34.4 | 5.9 | 13.5 | 14.9 |
| **Classical RLVR Methods** | | | | | | | |
| GRPO | 15.6 | 8.8 | 62.5 | 78.2 | 38.6 | 40.4 | 40.7 |
| SimpleRL-Zoo | 17.0 | 9.6 | 64.7 | 76.6 | 31.6 | 40.3 | 40.0 |
| OPO | 16.6 | 8.4 | 64.6 | 74.6 | 31.6 | 40.3 | 39.4 |
| **Scheduling Methods** | | | | | | | |
| $ACC_{sigmoid}$ | 16.7 | 9.8 | 68.6 | 79.0 | 34.2 | 39.4 | 41.3 |
| LPPO | 15.8 | 9.4 | 64.0 | 76.8 | 35.3 | 36.7 | 39.7 |
| Seed-GRPO | 13.3 | 6.0 | 63.3 | 76.6 | 32.4 | 36.3 | 38.0 |
| **Our Methods** | | | | | | | |
| Re-Schedule$_{linear}$ | **18.4** | 12.2 | 68.6 | 80.4 | 41.2 | 42.1 | 43.8 |
| Re-Schedule$_{sigmoid}$ | 18.2 | **14.0** | **69.2** | **81.0** | **41.5** | **43.3** | **44.5** |

## 7.3 ABLATION EXPERIMENT

Table 3: Ablation study on tree construction parameters. The default configuration (branching factor $k = 4$, depth $d = 4$) achieves the best performance.

| Branch $k$ | Depth $d$ | AIME24 | AIME25 | AMC23 | MATH500 | Minerva | Olympiad | Avg. |
|---|---|---|---|---|---|---|---|---|
| 4 | 4 | 34.2 | 16.0 | 71.1 | 81.8 | 42.3 | 44.4 | **48.3** |
| 3 | 5 | 33.8 | 14.8 | 68.4 | 79.6 | 42.3 | 42.8 | 46.9 |
| 5 | 3 | 31.7 | 14.2 | 70.4 | 81.0 | 41.9 | 43.0 | 47.0 |

We investigate the impact of the reasoning tree's structure by varying the branching factor $k$ and maximum depth $d$. The choice of these parameters determines the fidelity of the approximated reasoning tree. While larger values for $k$ and $d$ theoretically provide a more accurate approximation and thus a more effective r-score, they also introduce a significant computational overhead. As shown in Table 3, our default configuration of $k = 4$ and $d = 4$ yields the best average performance (48.3%). For more detailed analysis, please see Appendices D.3 and D.4.

Table 4: Ablation study on the weight function hyperparameters, $\omega_{min}$ and $\omega_{max}$. The default setting (0.5, 2.0) performs best.

| $\omega_{min}$ | $\omega_{max}$ | AIME24 | AIME25 | AMC23 | MATH500 | Minerva | Olympiad | Avg. |
|---|---|---|---|---|---|---|---|---|
| 0.5 | 2.0 | 35.2 | **16.0** | **72.3** | **82.2** | **42.3** | **44.4** | **48.5** |
| 0.5 | 1.5 | 31.4 | 15.4 | **72.3** | 81.8 | 38.1 | 42.5 | 46.9 |
| 0.5 | 3.0 | 33.5 | 15.0 | 69.1 | 81.8 | 37.5 | 41.0 | 46.3 |
| 0.8 | 2.0 | **36.6** | 13.6 | 71.1 | 81.6 | 37.1 | 43.8 | 47.3 |
| 0.2 | 2.0 | 33.5 | 13.9 | 71.0 | 80.0 | 38.2 | 41.6 | 46.4 |

We analyze the sensitivity of our method to the minimum $\omega_{min}$ and maximum $\omega_{max}$ weight hyperparameters, which control the dynamic range of the curriculum. Results in Table 4 show that our default setting of $\omega_{min} = 0.5$ and $\omega_{max} = 2.0$ achieves the highest average score (48.5). Decreasing the dynamic range by either reducing $\omega_{max}$ (to 1.5) or increasing $\omega_{min}$ (to 0.8) leads to performance degradation. This indicates that a sufficiently large weighting range is crucial for the curriculum to effectively differentiate between easy and hard samples. Conversely, an overly extreme range (e.g., $\omega_{min} = 0.2$) also degrades performance, possibly because the curriculum excessively under-weights

difficult queries. By assigning them a minimal weight for a prolonged period, the model is prevented from learning difficult queries. Furthermore, for additional experiments on the design choices for the r-score calculation, please see Appendix D.1.

Table 5: Computational cost vs. Performance gain. "Additional Cost" is relative to the total training time.

| Tree Size ($k^d$) | $3^3$ | $4^3$ | $3^4$ | $4^4$ (Default) |
|---|---|---|---|---|
| Time Cost (hours) | 3.48 | 6.21 | 6.70 | 22.67 |
| Additional Cost | +7.45% | +13.30% | +14.35% | +48.54% |
| Avg Performance Gain | +3.2 | +3.0 | +3.2 | +4.0 |

## 7.4 ANALYSIS EXPERIMENTS

### 7.4.1 COMPUTATIONAL COST ANALYSIS

As shown in Figure 5, we analyzed the trade-off between the offline tree construction cost and the resulting performance gain.

Table 5 presents the time cost measured on $8 \times$ H20 GPUs. While larger trees ($4^4$) incur higher preprocessing costs compared to smaller trees ($3^3$), the cost remains manageable relative to the total training time (approx. 46 hours for 5 epochs), and the performance gains are substantial.

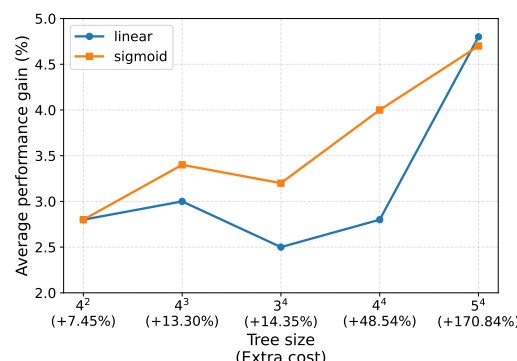

Figure 5: Average performance gain versus reasoning tree size and computational cost.

### 7.4.2 IMPACT OF ORDERING

Table 6: Comparison between Re-Schedule (Easy-to-Hard) and Reverse Schedule (Hard-to-Easy).

| Schedule | AIME24 | AIME25 | AMC23 | MATH500 | Minerva | Olympiad | Avg |
|---|---|---|---|---|---|---|---|
| *Linear Mapping* | | | | | | | |
| Re-Schedule (Ours) | 34.2 | 15.6 | 72.4 | 81.2 | 36.4 | 42.5 | 47.1 |
| Reverse Schedule | 31.9 | 14.0 | 67.6 | 81.0 | 37.8 | 41.8 | 45.7 |
| *Sigmoid Mapping* | | | | | | | |
| Re-Schedule (Ours) | 34.2 | 16.0 | 71.1 | 81.8 | 42.3 | 44.4 | 48.3 |
| Reverse Schedule | 30.2 | 15.4 | 67.1 | 80.6 | 34.9 | 40.2 | 44.7 |

To validate the "easy-to-hard" curriculum design, we compared our method against a "Reverse Schedule" where lower r-score (harder) samples are prioritized first. As shown in Table 6, the Reverse Schedule leads to a significant drop in performance, confirming that starting with structurally simple samples is crucial for effective learning.

## 8 CONCLUSIONS

In this work, we challenged the reliance on path-based metrics for RLVR data scheduling. We introduced the r-score, a novel metric that quantifies learnability based on the structure of a query's reasoning tree, and proposed Re-Schedule, a curriculum learning algorithm built upon it. Extensive experiments demonstrated that Re-Schedule consistently outperforms classical RLVR and existing scheduling methods, validating that r-score is a more effective proxy for learnability than path-based accuracy. Our findings establish that a structural understanding of the reasoning process provides a more powerful and principled foundation for creating efficient training curricula in RLVR.

ETHICS STATEMENT

We have manually reevaluated the dataset we created to ensure it is free of any potential for discrimination, human rights violations, bias, exploitation, and any other ethical concerns.

REPRODUCIBILITY STATEMENT

To ensure the reproducibility of our findings, all source code and datasets used in our experiments are included in the supplementary material. The provided materials are sufficient to replicate the main results presented in this paper.

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

## A   USAGE OF LLMS

Throughout the preparation of this manuscript, Large Language Models (LLMs) were utilized as a writing and editing tool. Specifically, we employed LLMs to improve the clarity and readability of the text, refine sentence structures, and correct grammatical errors. All final content, including the core scientific claims, experimental design, and conclusions, was conceived and written by us, and we take full responsibility for the final version of this paper.

## B   ACKNOWLEDGEMENTS

The authors would like to thank all the anonymous reviewers for their insightful comments and valuable suggestions. This research is supported by Anhui Provincial Natural Science Foundation (Grant No.2408085QF214), the Fundamental Research Funds for the Central Universities (Grant No.WK2080000206) and the Opening Project of the State Key Laboratory of General Artificial Intelligence (Project No.SKLAGI2025OP06). The work is also sponsored by the CodeBuddy (https://www.codebuddy.ai/). Special thanks go to Yi Liu and the team at Tencent for their generous support and assistance.

## C   DETAILS OF EXPERIMENTAL SETUP

All algorithms are implemented based on the official GRPO codebase within the VeRL framework. We use a learning rate of 1e-6 without warm-up across all experiments. At each rollout step, we generate 8 answers for each of 512 sampled questions, then split the data into 16 mini-batches and train the policy network for 16 gradient steps. Models are trained for at most 150 rollout steps. Unless otherwise specified, we follow GRPO's default design choices with token-level loss normalization without dynamic sampling and KL regularization. For all models, the maximum input length is 1024 and the minimum input length is 3072. All the experiments were conducted on H20 GPUs.

Note: The authors of Eurus-PRIME only published results from training on Qwen2.5-Math-7B. Therefore, we do not include results for the Qwen2.5-7B model in our comparison.

## D   SUPPLEMENTARY EXPERIMENT

### D.1   EFFECT OF METRIC SELECTION

Table 7: Ablation study comparing our proposed node-level modification metric ('Fix') with a branch-level 'Pruning' metric. 'Fix' consistently outperforms 'Pruning', validating our fine-grained approach.

| Model | AIME24 | AIME25 | AMC23 | MATH500 | Minerva | Olympiad | Avg. |
|---|---|---|---|---|---|---|---|
| Fix | | | | | | | |
| Re-Schedule$_{linear}$ | 34.2 | 15.6 | 72.4 | 81.2 | 36.4 | 42.5 | 47.1 |
| Re-Schedule$_{sigmoid}$ | 34.2 | 16.0 | 71.1 | 81.8 | 42.3 | 44.4 | 48.3 |
| Pruning | | | | | | | |
| Re-Schedule$_{linear}$ | 35.7 | 14.6 | 73.7 | 81.0 | 34.9 | 41.6 | 46.9 |
| Re-Schedule$_{sigmoid}$ | 33.1 | 16.7 | 71.1 | 82.0 | 39.0 | 42.4 | 47.4 |

We validate our core design choice for the r-score calculation. Our proposed method ('Fix') defines an 'edit' as a single node modification. We compare this against an alternative ('Pruning'), where an 'edit' is defined as pruning an entire sub-branch from a decision point. Table 7 shows that the 'Fix' method consistently outperforms 'Pruning' for both linear (47.1% vs. 46.9%) and sigmoid (48.3% vs. 47.4%) schedules. This result shows that compared with the branch 'Pruning', the node 'Fix' is more consistent with the training process of reinforcement learning.

Table 8: Variance of Re-Schedule over multiple runs.

| | AIME24 | AIME25 | AMC23 | MATH500 | Minerva | Olympiad |
|---|---|---|---|---|---|---|
| Avg. $\pm$ Var. | $35.2 \pm 0.1$ | $16.0 \pm 0.0$ | $72.3 \pm 0.4$ | $82.2 \pm 0.0$ | $42.3 \pm 0.6$ | $44.4 \pm 0.6$ |

## D.2 VARIANCE ANALYSIS

To assess the stability of our proposed Re-Schedule method, we conducted repeated experiments using different random seeds. Table 8 reports the variance. The results demonstrate that our method exhibits low variance. This confirms that the reported improvements are statistically stable and not due to random variation. Compared to performance improvements, the impact of variance is minimal.

## D.3 ABLATION ON TOKEN INTERVAL $l$

To investigate the impact of $l$ on performance and its relationship with model capabilities, we conducted ablation studies on two base models: Qwen2.5-Math-7B and Qwen2.5-7B. We utilized the Sigmoid weighting mapping for these experiments.

Table 9 summarizes the average accuracy across six benchmarks (AIME24, AIME25, AMC23, MATH500, Minerva, Olympiad) for varying token intervals $l \in \{200, 400, 600, 1200\}$.

Table 9: Impact of Token Interval $l$ on Average Accuracy (Sigmoid Mapping).

| Interval ($l$) | Qwen2.5-Math-7B (Avg) | Qwen2.5-7B (Avg) |
|---|---|---|
| $l = 200$ | 48.3 | 44.5 |
| $l = 400$ | 48.6 | 43.2 |
| $l = 600$ | 48.9 | 43.1 |
| $l = 1200$ | 46.0 | 41.1 |

The results indicate that Re-Schedule is generally robust to the choice of $l$ within the range of $[200, 600]$. For the specialized math model (Qwen2.5-Math-7B), performance remains high and stable as $l$ increases to 600. For the general model (Qwen2.5-7B), while $l = 200$ yields the best results, the performance drop at $l = 600$ is relatively contained. A significant performance drop is observed for both models when $l = 1200$. This suggests that when the interval is too large, the approximated reasoning tree becomes too coarse to capture the critical branching points necessary for effective r-score estimation.

## D.4 SENSITIVITY TO BRANCHING FACTOR $k$, DEPTH $d$ AND MODIFICATION BUDGET $M$

We investigated the impact of the reasoning tree size on performance by varying the branching factor $k$ and depth $d$ on the Qwen2.5-Math-7B model.

Table 10: Ablation study on branching factor $k$ (with fixed $d = 4$).

| Setting | AIME24 | AIME25 | AMC23 | MATH500 | Minerva | Olympiad | Avg |
|---|---|---|---|---|---|---|---|
| *Linear Mapping* | | | | | | | |
| $k = 3$ | 32.5 | 15.2 | 74.0 | 81.8 | 36.3 | 41.0 | 46.8 |
| $k = 4$ (Default) | 34.2 | 15.6 | 72.4 | 81.2 | 36.4 | 42.5 | 47.1 |
| $k = 5$ | 35.1 | 18.1 | 77.4 | 81.8 | 37.8 | 44.6 | 49.1 |
| *Sigmoid Mapping* | | | | | | | |
| $k = 3$ | 32.0 | 15.2 | 74.2 | 81.6 | 38.6 | 43.1 | 47.5 |
| $k = 4$ (Default) | 34.2 | 16.0 | 71.1 | 81.8 | 42.3 | 44.4 | 48.3 |
| $k = 5$ | 36.4 | 17.0 | 75.5 | 81.2 | 40.6 | 43.4 | 49.0 |

Table 11: Ablation study on tree depth $d$ (with fixed $k = 4$).

| Setting | AIME24 | AIME25 | AMC23 | MATH500 | Minerva | Olympiad | Avg |
|---|---|---|---|---|---|---|---|
| | | | *Linear Mapping* | | | | |
| $d = 2$ | 31.1 | 15.3 | 74.2 | 81.8 | 38.2 | 42.2 | 47.1 |
| $d = 3$ | 31.4 | 14.6 | 72.7 | 82.0 | 39.7 | 43.3 | 47.3 |
| $d = 4$ (Default) | 34.2 | 15.6 | 72.4 | 81.2 | 36.4 | 42.5 | 47.1 |
| | | | *Sigmoid Mapping* | | | | |
| $d = 2$ | 31.9 | 14.8 | 74.5 | 81.6 | 37.2 | 42.4 | 47.1 |
| $d = 3$ | 33.2 | 16.4 | 73.0 | 80.0 | 41.9 | 41.6 | 47.7 |
| $d = 4$ (Default) | 34.2 | 16.0 | 71.1 | 81.8 | 42.3 | 44.4 | 48.3 |

Table 12: Ablation study on node modification budget $M$ (with fixed $k = 4, d = 4, l = 200$).

| Setting | AIME24 | AIME25 | AMC23 | MATH500 | Minerva | Olympiad | Avg |
|---|---|---|---|---|---|---|---|
| $M = 5$ | 33.6 | 15.4 | 72.2 | 79.0 | 40.4 | 43.1 | 47.3 |
| $M = 10$ (Default) | 34.2 | 16.0 | 71.1 | 81.8 | 42.3 | 44.4 | **48.3** |
| $M = 15$ | 34.7 | 16.0 | 71.8 | 82.0 | 41.6 | 42.4 | 48.1 |

**Varying branching factor** $k$**:** Fixing $d = 4$ and $l = 200$, we tested $k \in \{3, 4, 5\}$. As shown in Table 10, increasing $k$ generally improves performance, suggesting that a denser tree captures the structural difficulty more accurately.

**Varying tree depth** $d$**:** Fixing $k = 4$ and $l = 200$, we tested $d \in \{2, 3, 4\}$. Table 11 shows that deeper trees provide a better estimation of the reasoning structure, leading to improved downstream performance.

**Varying node modification budget** $M$**:** Finally, we assess the stability of our method with respect to the node modification budget $M$. Fixing $k = 4, d = 4$, and $l = 200$, we evaluated performance across $M \in \{5, 10, 15\}$. As presented in Table 12, the results are relatively robust to changes in this parameter. While the default setting of $M = 10$ yields the optimal average accuracy, varying the budget between 5 and 15 results in no significant performance degradation, indicating that the r-score remains a reliable metric across different budget constraints.

### D.5 GENERALIZATION TO DIFFERENT MODEL ARCHITECTURES

To demonstrate the broad applicability of our method beyond the Qwen2.5 family, we conducted additional experiments on the Qwen3-4B-Base model.

Table 13: Performance comparison on Qwen3-4B-Base.

| Model | AIME24 | AIME25 | AMC23 | MATH500 | Minerva | Olympiad | Avg |
|---|---|---|---|---|---|---|---|
| GRPO | 24.2 | 21.8 | 52.4 | 86.0 | 39.4 | 43.4 | 44.5 |
| ACC | 24.8 | 23.3 | 59.3 | 88.8 | 41.6 | 42.0 | 46.6 |
| Re-Schedule (Ours) | 27.6 | 26.9 | 57.6 | 89.8 | 43.5 | 47.4 | 48.8 |

As shown in Table 13, Re-Schedule consistently outperforms both the standard GRPO baseline and the accuracy-based curriculum (ACC) across all benchmarks. This confirms that the effectiveness of the r-score is not limited to specific model architectures or sizes.

### D.6 DYNAMIC R-SCORE CALCULATION

To determine if the r-score should be updated as the model evolves, we compared our standard static approach (computed once before training) with a dynamic approach where the r-score is re-computed and weights are updated three times during the training process.

Table 14: Comparison between Static and Dynamic R-Score updates on Qwen2.5-Math-7B.

| Method | AIME24 | AIME25 | AMC23 | MATH500 | Minerva | Olympiad | Avg |
|---|---|---|---|---|---|---|---|
| *Linear Mapping* | | | | | | | |
| Static (Default) | 34.2 | 15.6 | 72.4 | 81.2 | 36.4 | 42.5 | 47.1 |
| Dynamic (3 updates) | 34.6 | 14.9 | 75.3 | 80.0 | 39.7 | 41.6 | 47.8 |
| *Sigmoid Mapping* | | | | | | | |
| Static (Default) | 34.2 | 16.0 | 71.1 | 81.8 | 42.3 | 44.4 | 48.3 |
| Dynamic (3 updates) | 35.3 | 15.2 | 74.2 | 82.4 | 42.7 | 43.6 | 48.9 |

As shown in Table 14, the dynamic approach yields performance comparable to the static baseline (e.g., 48.9% vs. 48.3% for Sigmoid mapping). Given the substantial computational cost of re-generating reasoning trees during training, we conclude that the static r-score serves as a sufficient and efficient prior for guiding the curriculum.

### D.7 GENERALIZATION TO CODE GENERATION

Table 15: Performance comparison on Code Generation (LiveCodeBench v5).

| Method | pass@1 | pass@4 |
|---|---|---|
| GRPO | 25.4 | 35.4 |
| $ACC_{sigmoid}$ | 25.8 | 36.0 |
| Re-Schedule$_{sigmoid}$ | 26.3 | 37.8 |

To validate the generalization capability of Re-Schedule beyond mathematical reasoning, we extended our evaluation to the domain of Code Generation. We utilized DeepSeek-R1-Distill-Qwen-1.5B as the base model. The model was trained on the ArcherCodeR dataset Wang et al. (2025d), which contains 6,753 code generation tasks. For evaluation, we used the LiveCodeBench v5 benchmark Jain et al. (2024). We report pass@1 and pass@4 metrics (averaged over 8 samples).

As shown in Table 15, Re-Schedule consistently outperforms both the standard GRPO baseline and the accuracy-based curriculum (ACC). Specifically, our method achieves a +0.9% improvement in pass@1 and a significant +2.4% improvement in pass@4 compared to GRPO. These results confirm that the structural insights captured by the r-score are effective in the coding domain, where the "reasoning tree" corresponds to the decision space of code logic and syntax. Looking forward, the tree-based structural metrics could also be adapted to improve multi-modal data generation pipelines and self-instructed compositional code captioning (Wang et al., 2024; 2025b;c; Hao et al., 2026; Wang et al., 2025a; Lei et al., 2025).

