# OpenReview forum: "Scheduling Your LLM Reinforcement Learning with Reasoning Trees"
_ICLR.cc/2026/Conference — ICLR 2026 Poster_

### Official Review · Reviewer_eZz4 · 2025-10-29

**Soundness:** 3
**Presentation:** 3
**Contribution:** 3
**Rating:** 6
**Confidence:** 4

**Summary:**

This paper introduces R-Score, a novel metric to quantify the learnability of queries in RL. R-Score measures how many node modifications are required in a reasoning tree to achieve a correct answer, reflecting how suitable a sample is for training.
In experiments, curriculum learning based on R-Score achieves better performance than accuracy-based and random scheduling methods, demonstrating that R-Score can effectively guide sample selection during RL training.

**Strengths:**

1. The paper proposes a new metric, R-Score, to evaluate whether a question is appropriate for RL training. The design aligns well with the intuition that queries that require fewer reasoning corrections are easier to learn, and the empirical results consistently support this idea.

2. The paper is clearly written, with detailed explanations of the R-Score computation, experimental setup, and ablation studies. Figures and tables effectively illustrate the trends and improvements.

**Weaknesses:**

1. The method is only validated on the Qwen2.5 and math task, which is a special combination that makes the results less general [1]. To establish broader applicability, more models and diverse reasoning tasks should be included.

2. The computation of R-Score depends heavily on several hyperparameters such as the branching factor, depth, and node modification budget. These choices can dramatically increase computational requirements, making large-scale use impractical without further optimization or approximation.

[1] Wu, Mingqi, et al. "Reasoning or memorization? unreliable results of reinforcement learning due to data contamination." arXiv preprint arXiv:2507.10532 (2025).

**Questions:**

Is the R-Score static during training, or do you recompute it as the model improves?
Since the model’s competence evolves, a query that was previously hard may later become easy. Would it be beneficial or necessary to periodically update the R-Score to reflect this changing learnability?

---

> ### Author Response · Authors · 2025-11-24
> **Response to Reviewer eZz4 (1/3)**
>
> We sincerely thank you for your time and the constructive feedback. We are encouraged by your recognition of R-Score as a novel and intuitive metric, and your appreciation of our paper’s clarity and empirical results. We have carefully considered your comments regarding generalization, hyperparameter sensitivity, and the dynamic nature of the metric. Below, we provide detailed responses and new experimental evidence to address your concerns.
>
> ---
>
> ## Weakness 1: Generalization to Other Models and Domains
>
> > The method is only validated on the Qwen2.5 and math task, which is a special combination that makes the results less general [1]. To establish broader applicability, more models and diverse reasoning tasks should be included.
>
> Thank you for this valuable suggestion. We initially focused on mathematical reasoning because it offers clear, verifiable rewards, making it an ideal testbed for RLVR methods. However, we fully agree that demonstrating broader applicability is crucial.
>
> 1. Validation on the Additional Model To address the concern regarding model specificity, we conducted additional experiments on Qwen3-4B-Base. As shown in the table below, Re-Schedule consistently outperforms baselines, confirming that our method's effectiveness is not limited to a specific model architecture or size.  These results have been added to **Appendix C.5**.
>
> | Model: Qwen3-4B-Base | AIME24 | AIME25 | AMC23 | MATH500 | Minerva | Olympiad | Avg  |
> | -------------------- | ------ | ------ | ----- | ------- | ------- | -------- | ---- |
> | GRPO                 | 24.2   | 21.8   | 52.4  | 86      | 39.4    | 43.4     | 44.5 |
> | ACC                  | 24.8   | 23.3   | 59.3  | 88.8    | 41.6    | 42       | 46.6 |
> | Re-Schedule (Ours)   | 27.6   | 26.9   | 57.6  | 89.8    | 43.5    | 47.4     | 48.8 |
>
> 1. Generalization to Other Domains (Code Generation) We believe the core philosophy of the r-score holds significant potential for other reasoning-intensive domains, such as Code Generation:
>    1. Reasoning Tree: Represents the space of potential code implementation paths.
>    2. Nodes: Represent code blocks or logical decisions.
>    3. Rewards: Verifiable via unit tests.
>    4. r-score (Structural Complexity): Measures the "repair potential." A high r-score implies a structurally simple bug (fixable via limited edits), whereas a low r-score implies a need for complex algorithmic refactoring.
>
> To empirically validate this, we conducted supplementary experiments on the **Code Generation task**. We utilized DeepSeek-R1-Distill-Qwen-1.5B as the base model, training on the ArcherCodeR dataset (6,753 samples) and evaluating on LiveCodeBench v5.
>
> As shown in the table below, Re-Schedule consistently outperforms both the standard GRPO baseline and the accuracy-based curriculum (ACC).
>
> | Method                | pass@1 | pass@4 |
> | --------------------- | ------ | ------ |
> | GRPO                  | 25.4   | 35.4   |
> | ACC (Sigmoid)         | 25.8   | 36     |
> | Re-Schedule (Sigmoid) | 26.3   | 37.8   |
>
> Our method achieves a +0.9% gain in pass@1 and a significant +2.4% gain in pass@4 compared to GRPO. Notably, it also surpasses the accuracy-based curriculum (ACC), confirming that the r-score captures the structural difficulty of coding problems more effectively than simple accuracy rates. These results demonstrate that the Re-Schedule framework generalizes well to the code generation domain.
>
> These experimental content has been added to **Appendix C.7**.

---

> ### Author Response · Authors · 2025-11-24
> **Response to Reviewer eZz4 (2/3)**
>
> ## Weakness 2: Hyperparameter Sensitivity and Computational Cost
>
> > The computation of R-Score depends heavily on several hyperparameters such as the branching factor, depth, and node modification budget. These choices can dramatically increase computational requirements, making large-scale use impractical without further optimization or approximation.
>
> Thank you for highlighting the computational overhead to build approximate reasoning trees. In fact, it does not pose a substantial computational burden. Moreover, we can reduce the cost by shrinking the tree via smaller values of $k$ and $d$. The relationship between the additional computational overhead and the parameters $k$ and $d$ is shown in the table below.
>
> **Computational Cost Analysis** We measured the offline construction time on 8 $\times$ H20 GPUs. For context, standard RL training takes approximately 9.34 hours per epoch (typically 5 epochs total).
>
> | Tree size $k^d$          | $3^3$  | $4^3$   | $3^4$   | $4^4$   |
> | ------------------------ | ------ | ------- | ------- | ------- |
> | Time cost (hours)        | 3.48   | 6.21    | 6.70    | 22.67   |
> | Additional time cost     | +7.45% | +13.30% | +14.35% | +48.54% |
> | Average performance gain | +3.2   | +3.0    | +3.2    | +4.0    |
>
> Additional time cost refers to the additional time overhead when constructing reasoning trees compared to the training process.
>
> This analysis confirms that while larger trees increase preprocessing time, the cost remains feasible, and researchers can select parameters based on their available resources. For example, under tight computational budgets, setting ($k=3$) and ($d=3$) incurs about $7%$ additional overhead relative to the baseline method, yet yields an average improvement of $3.2$, etc. In sum, we observe a clear trade-off: practitioners can flexibly choose the tree size according to available compute to realize performance gains.
>
> Besides, we performed extensive ablation studies (varying branching factor $k$, depth $d$). Our findings provide a principled guide for these settings:
>
> 1. **The Trade-off Principle: Performance vs. Overhead** There is a direct trade-off: **larger trees yield better performance but incur higher costs.**
>    1. **Performance:** Increasing $k$ or $d$ creates a finer-grained approximation of the true reasoning space, allowing for a more accurate r-score estimation and better downstream accuracy (as seen in the tables below).
>    2. **Overhead:** The offline construction cost grows with tree size ($k^d$). However, by utilizing **KV-Cache reuse**, we significantly mitigate redundant computations.
>    3. **Recommendation:** Our default setting ($k=4, d=4$) represents a "sweet spot" that balances high performance with manageable offline overhead (far less than the RL training cost itself). Researchers with abundant resources can increase $k, d$ for marginal gains, while those with limited resources can use smaller trees and still achieve improvements.
> 2. **Supplementary Experiments** We conducted parametric analysis experiments on $k$, $d$ in **Section 7.3** of the original paper. Furthermore, during the rebuttal, we performed additional experiments on Qwen2.5-Math-7B to further verify these claims.
>
> **(a) Varying Branching Factor ($k$)** (Fixed $d=4, l=200$)
>
> | Setting         | AIME24 | AIME25 | AMC23 | MATH500 | Minerva | Olympiad | Avg  |
> | --------------- | ------ | ------ | ----- | ------- | ------- | -------- | ---- |
> | Linear Mapping  |        |        |       |         |         |          |      |
> | $k=3$           | 32.5   | 15.2   | 74    | 81.8    | 36.3    | 41       | 46.8 |
> | $k=4$ (Default) | 34.2   | 15.6   | 72.4  | 81.2    | 36.4    | 42.5     | 47.1 |
> | $k=5$           | 35.1   | 18.1   | 77.4  | 81.8    | 37.8    | 44.6     | 49.1 |
> | Sigmoid Mapping |        |        |       |         |         |          |      |
> | $k=3$           | 32     | 15.2   | 74.2  | 81.6    | 38.6    | 43.1     | 47.5 |
> | $k=4$ (Default) | 34.2   | 16     | 71.1  | 81.8    | 42.3    | 44.4     | 48.3 |
> | $k=5$           | 36.4   | 17     | 75.5  | 81.2    | 40.6    | 43.4     | 49   |

---

> ### Author Response · Authors · 2025-11-24
> **Response to Reviewer eZz4 (3/3)**
>
> **(b) Varying** **Tree** **Depth ($d$)** (Fixed $k=4, l=200$)
>
> | Setting         | AIME24 | AIME25 | AMC23 | MATH500 | Minerva | Olympiad | Avg  |
> | --------------- | ------ | ------ | ----- | ------- | ------- | -------- | ---- |
> | Linear Mapping  |        |        |       |         |         |          |      |
> | $d=2$           | 31.1   | 15.3   | 74.2  | 81.8    | 38.2    | 42.2     | 47.1 |
> | $d=3$           | 31.4   | 14.6   | 72.7  | 82      | 39.7    | 43.3     | 47.3 |
> | $d=4$ (Default) | 34.2   | 15.6   | 72.4  | 81.2    | 36.4    | 42.5     | 47.1 |
> | Sigmoid Mapping |        |        |       |         |         |          |      |
> | $d=2$           | 31.9   | 14.8   | 74.5  | 81.6    | 37.2    | 42.4     | 47.1 |
> | $d=3$           | 33.2   | 16.4   | 73    | 80      | 41.9    | 41.6     | 47.7 |
> | $d=4$ (Default) | 34.2   | 16     | 71.1  | 81.8    | 42.3    | 44.4     | 48.3 |
>
> **(c) Node modification budget ($M$)** (Fixed $k=4, d=4, l=200$)
>
> | Setting          | AIME24 | AIME25 | AMC23 | MATH500 | Minerva | Olympiad | Avg  |
> | ---------------- | ------ | ------ | ----- | ------- | ------- | -------- | ---- |
> | $M=5$            | 33.6   | 15.4   | 72.2  | 79      | 40.4    | 43.1     | 47.3 |
> | $M=10$ (Default) | 34.2   | 16     | 71.1  | 81.8    | 42.3    | 44.4     | 48.3 |
> | $M=15$           | 34.7   | 16     | 71.8  | 82      | 41.6    | 42.4     | 48.1 |
>
> We also conduct ablation studies to assess the stability of the results with respect to the Node modification budget. In practice, setting this budget between 5 and 15 had no significant impact on performance.
>
> These experimental content has been added to **Appendix C.4 and Section 7.4.1**.
>
> ## Question 1: Static vs. Dynamic R-Score
>
> > Is the R-Score static during training, or do you recompute it as the model improves? Since the model’s competence evolves, a query that was previously hard may later become easy. Would it be beneficial or necessary to periodically update the R-Score to reflect this changing learnability?
>
> To empirically verify this design choice, we conducted an experiment where we re-computed the R-Score and updated the weighting strategy three times during the training process.
>
> | Model: Qwen2.5-Math-7B            | AIME24 | AIME25 | AMC23 | MATH500 | Minerva | Olympiad | Avg  |
> | --------------------------------- | ------ | ------ | ----- | ------- | ------- | -------- | ---- |
> | Linear Mapping                    |        |        |       |         |         |          |      |
> | Re-Schedule (Static)              | 34.2   | 15.6   | 72.4  | 81.2    | 36.4    | 42.5     | 47.1 |
> | Re-Schedule (Dynamic - 3 updates) | 34.6   | 14.9   | 75.3  | 80      | 39.7    | 41.6     | 47.8 |
> | Sigmoid Mapping                   |        |        |       |         |         |          |      |
> | Re-Schedule (Static)              | 34.2   | 16     | 71.1  | 81.8    | 42.3    | 44.4     | 48.3 |
> | Re-Schedule (Dynamic - 3 updates) | 35.3   | 15.2   | 74.2  | 82.4    | 42.7    | 43.6     | 48.9 |
>
> Conclusion: The dynamic approach yields comparable performance to the static baseline (e.g., 48.9 vs. 48.3). Given the substantial computational cost of re-generating reasoning trees during training, we conclude that the static R-Score serves as a sufficient and efficient prior for guiding the curriculum. These results have been added to **Appendix C.6**.
>
> ---
>
> ## **Thanks again**
>
> We sincerely thank you again for your constructive and detailed feedback, which has helped us identify clear pathways to improve our paper. Should you have any further questions or require additional discussion, please don't hesitate to reach out. If we have adequately addressed your concerns, we would be grateful for your consideration in adjusting your evaluation score accordingly.

---

### Official Review · Reviewer_Hh8P · 2025-10-30

**Soundness:** 3
**Presentation:** 3
**Contribution:** 2
**Rating:** 6
**Confidence:** 3

**Summary:**

This paper challenges the use of path-based accuracy for RLVR data scheduling, arguing it is a poor proxy for true learning difficulty. The authors reframe the problem around the "Reasoning Tree" structure, proposing the "r-score": a novel metric that quantifies a query's "learning potential" based on the maximum accuracy gain achievable under a limited "node editing budget". They introduce "Re-Schedule", a curriculum learning algorithm that uses the r-score to prioritize structurally simple queries first. Experiments on six math benchmarks show that Re-Schedule significantly outperforms strong baselines, achieving state-of-the-art results.

**Strengths:**

1.  Insightful Problem Definition: The paper provides a valuable reframing of the scheduling problem, moving from simple accuracy to the more nuanced "structural complexity" of a query's reasoning tree. The "node-editing" analogy is highly intuitive.

2.  Principled Metric (r-score): The r-score is a well-defined metric that attempts to directly quantify "learning potential" rather than just static difficulty. This is a more principled approach to capturing the learning dynamics of a query.

3.  Strong Empirical Results: The Re-Schedule method achieves state-of-the-art results across six math benchmarks. The significant accuracy gains (up to 3.8%) over strong RLVR and scheduling baselines compellingly demonstrate the method's effectiveness.

**Weaknesses:**

1.  Limited Experimental Domain: The method's effectiveness is only validated on math-reasoning tasks. It is unclear if the r-score framework will generalize to other domains. Code generation would be a good benchmark with verifiable rewards.

2.  Insufficient Discussion of Related Work: The paper needs to better position its contributions relative to the broader curriculum learning (CL) field. A deeper comparison with other methods that also estimate 'learning potential' (e.g., via gradients or uncertainty) is warranted.

**Questions:**

3.  High Computational Overhead: The method relies on a substantial offline pre-computation stage to build approximate trees and calculate r-scores for all queries, which may be computationally infeasible for very large-scale datasets.

4.  Hyperparameter Sensitivity: The r-score's quality depends on tree approximation hyperparameters ($k$, $d$, $w$). As Table 3 and Table 4 suggests, performance is sensitive to these choices, but the paper lacks a principled guide for setting them.

---

> ### Author Response · Authors · 2025-11-24
> **Response to Reviewer Hh8P (1/3)**
>
> We sincerely thank the reviewer for the insightful feedback and for recognizing the value of our "Reasoning Tree" formulation and the "r-score" metric. We appreciate the opportunity to clarify the generalizability of our method, the robustness of our hyperparameters, and the computational trade-offs. We have addressed your concerns point-by-point below.
>
> ## Weakness 1: Generalization to Other Domains
>
> > Limited Experimental Domain: The method's effectiveness is only validated on math-reasoning tasks. It is unclear if the r-score framework will generalize to other domains. Code generation would be a good benchmark with verifiable rewards.
>
> Thank you for this insightful suggestion. We conducted supplementary experiments on the Code Generation task. We use DeepSeek-R1-Distill-Qwen-1.5B as the base model for stable performance, training on the ArcherCodeR dataset (6,753 samples) and evaluating on LiveCodeBench v5.
>
> As shown in the table below, Re-Schedule consistently outperforms both the standard GRPO baseline and the accuracy-based curriculum (ACC).
>
> | Method                | pass@1 | pass@4 |
> | --------------------- | ------ | ------ |
> | GRPO                  | 25.4   | 35.4   |
> | ACC (Sigmoid)         | 25.8   | 36     |
> | Re-Schedule (Sigmoid) | 26.3   | 37.8   |
>
> Our method achieves a +0.9% gain in pass@1 and a significant +2.4% gain in pass@4 compared to GRPO. Notably, it also surpasses the accuracy-based curriculum (ACC), confirming that the r-score captures the structural difficulty of coding problems more effectively than simple accuracy rates. These results demonstrate that the Re-Schedule framework generalizes well to the code generation domain.
>
> These experimental content has been added to **Appendix C.7**.
>
> ## Weakness 2: Discussion of Related Work
>
> > Insufficient Discussion of Related Work: The paper needs to better position its contributions relative to the broader curriculum learning (CL) field. A deeper comparison with other methods that also estimate 'learning potential' (e.g., via gradients or uncertainty) is warranted.
>
> We appreciate this suggestion. We have revised **Section 2.2** to better position our work within the CL field.
>
> The core distinction of our approach lies in how we model the problem space. LLM reasoning tasks inherently possess a tree-structured solution space (ReasoningTree). While existing methods operate on this space, they differ fundamentally in how they estimate difficulty:
>
> - **Existing methods** (e.g., LPPO [1], Seed-GRPO [2]) rely on **scalar proxies**—such as accuracy gradients or semantic uncertainty—to plan the curriculum. These methods effectively treat the reasoning process as a flat sequence, overlooking the topological relationship between reasoning steps.
> - **Our method (Re-Schedule)** explicitly leverages the **inherent structure** of the reasoning tree. Instead of relying on indirect proxies, we calculate the *Reasoning Score* based on the actual topology of solution paths (i.e., how correct and incorrect paths branch and diverge).
>
>
> This structural awareness allows Re-Schedule to identify "high-potential" queries that accuracy-based or uncertainty-based metrics often misclassify, thereby achieving superior performance.
>
>
> **References：**
>
> [1] From Data-Centric to Sample-Centric: Enhancing LLM Reasoning via Progressive Optimization, 2025
>
> [2] Seed-grpo: Semantic entropy enhanced grpo for uncertainty-aware policy optimization, 2025

---

> ### Author Response · Authors · 2025-11-24
> **Response to Reviewer Hh8P (2/3)**
>
> ## Question 1: Computational Overhead
>
> > High Computational Overhead: The method relies on a substantial offline pre-computation stage to build approximate trees and calculate r-scores for all queries, which may be computationally infeasible for very large-scale datasets.
>
> Thank you for highlighting the computational overhead to build approximate reasoning trees. In fact, it does not pose a substantial computational burden. Moreover, we can reduce the cost by shrinking the tree via smaller values of $k$ and $d$. The relationship between the additional computational overhead and the parameters kkk and ddd is shown in the table below.
>
> **Computational Cost Analysis** We measured the offline construction time on 8 $\times$ H20 GPUs. For context, standard RL training takes approximately 9.34 hours per epoch (typically 5 epochs total).
>
> | Tree size $k^d$          | $3^3$  | $4^3$   | $3^4$   | $4^4$   |
> | ------------------------ | ------ | ------- | ------- | ------- |
> | Time cost (hours)        | 3.48   | 6.21    | 6.70    | 22.67   |
> | Additional time cost     | +7.45% | +13.30% | +14.35% | +48.54% |
> | Average performance gain | +3.2   | +3.0    | +3.2    | +4.0    |
>
> Additional time cost refers to the additional time overhead when constructing reasoning trees compared to the training process.
>
> We can observe a trade-off between computational cost and hyperparameter selection. This results confirms that while larger trees increase preprocessing time, the cost remains feasible, and researchers can select parameters based on their available resources. For example, under tight computational budgets, setting ($k=3$) and ($d=3$) incurs about $7%$ additional overhead relative to the baseline method, yet yields an average improvement of $3.2$, etc. In sum, we observe a clear trade-off: practitioners can flexibly choose the tree size according to available compute to realize performance gains.
>
> These experimental content has been added to **Section 7.4.1**.

---

> ### Author Response · Authors · 2025-11-24
> **Response to Reviewer Hh8P (3/3)**
>
> ## Question 2: Hyperparameter Sensitivity
>
> > Hyperparameter Sensitivity: The r-score's quality depends on tree approximation hyperparameters ($k$, $d$, $w$). As Table 3 and Table 4 suggests, performance is sensitive to these choices, but the paper lacks a principled guide for setting them.
>
> To address your concerns regarding the sensitivity and scalability of our tree construction parameters, we conducted an additional ablation study. Our results demonstrate the robustness of parameter $l$, while clarifying that parameters $k$ and $d$ represent a trade-off between performance and computational cost.
>
> We conducted parametric analysis experiments on $k$, $d$ in **Section 7.3** of the original paper. Furthermore, during the rebuttal, we performed additional experiments on Qwen2.5-Math-7B to further verify these claims.
>
> As for $\omega$, the hyperparameter sensitivity analysis on the weight function hyperparameters has been shown in the **ablation study Table 4**.
>
> - **The Performance-Cost Trade-off ($k$ and  $d$)** The parameters $k$ (branching factor) and $d$ (depth) define an approximation of the true, exponentially large reasoning tree.
>   - **Performance:** As shown in the tables below, increasing $k$ or $d$ generally leads to better performance. A finer-grained tree captures structural difficulty more faithfully, yielding a more accurate r-score estimate; this, in turn, demonstrates that with sufficient compute, our method holds strong promise.
>   - **Cost:** As mentioned in the response to Question 1, this improvement has computational costs.. While our KV-Cache reuse strategy significantly reduces redundancy, the sampling cost grows with $k^d$.
>   - **Conclusion:** Our default setting ($k=4, d=4$) represents a "sweet spot" that balances high performance with manageable offline preprocessing costs. However, for scenarios with abundant compute, larger trees can unlock further gains.
>
> **(a) Varying Branching Factor ($k$)** (Fixed $d=4, l=200$)
>
> | Setting         | AIME24 | AIME25 | AMC23 | MATH500 | Minerva | Olympiad | Avg  |
> | --------------- | ------ | ------ | ----- | ------- | ------- | -------- | ---- |
> | Linear Mapping  |        |        |       |         |         |          |      |
> | $k=3$           | 32.5   | 15.2   | 74    | 81.8    | 36.3    | 41       | 46.8 |
> | $k=4$ (Default) | 34.2   | 15.6   | 72.4  | 81.2    | 36.4    | 42.5     | 47.1 |
> | $k=5$           | 35.1   | 18.1   | 77.4  | 81.8    | 37.8    | 44.6     | 49.1 |
> | Sigmoid Mapping |        |        |       |         |         |          |      |
> | $k=3$           | 32     | 15.2   | 74.2  | 81.6    | 38.6    | 43.1     | 47.5 |
> | $k=4$ (Default) | 34.2   | 16     | 71.1  | 81.8    | 42.3    | 44.4     | 48.3 |
> | $k=5$           | 36.4   | 17     | 75.5  | 81.2    | 40.6    | 43.4     | 49   |
>
> **(b) Varying** **Tree** **Depth ($d$)** (Fixed $k=4, l=200$)
>
> | Setting         | AIME24 | AIME25 | AMC23 | MATH500 | Minerva | Olympiad | Avg  |
> | --------------- | ------ | ------ | ----- | ------- | ------- | -------- | ---- |
> | Linear Mapping  |        |        |       |         |         |          |      |
> | $d=2$           | 31.1   | 15.3   | 74.2  | 81.8    | 38.2    | 42.2     | 47.1 |
> | $d=3$           | 31.4   | 14.6   | 72.7  | 82      | 39.7    | 43.3     | 47.3 |
> | $d=4$ (Default) | 34.2   | 15.6   | 72.4  | 81.2    | 36.4    | 42.5     | 47.1 |
> | Sigmoid Mapping |        |        |       |         |         |          |      |
> | $d=2$           | 31.9   | 14.8   | 74.5  | 81.6    | 37.2    | 42.4     | 47.1 |
> | $d=3$           | 33.2   | 16.4   | 73    | 80      | 41.9    | 41.6     | 47.7 |
> | $d=4$ (Default) | 34.2   | 16     | 71.1  | 81.8    | 42.3    | 44.4     | 48.3 |
>
> These experimental content has been added to **Appendix C.4**.
>
> ---
>
> ## **Thanks again**
>
> We sincerely thank you again for your constructive and detailed feedback, which has helped us identify clear pathways to improve our paper. Should you have any further questions or require additional discussion, please don't hesitate to reach out. If we have adequately addressed your concerns, we would be grateful for your consideration in adjusting your evaluation score accordingly.

---

> ### Comment · Reviewer_Hh8P · 2025-11-28
>
> Thanks for your response, and the revision resolved my concerns.
> I have no further questions.

---

### Official Review · Reviewer_qoyQ · 2025-10-31

**Soundness:** 3
**Presentation:** 2
**Contribution:** 2
**Rating:** 4
**Confidence:** 3

**Summary:**

This paper focuses on designing a more effective data scheduling strategy that leverages the structural relationships among data samples through reasoning trees. The proposed R-score captures these internal structural dependencies, while traditional path-based metrics such as accuracy fail to do so. Building on this metric, the Re-Schedule algorithm prioritizes high R-score samples early in training, using curriculum learning principles to balance exploration and exploitation and enhance generalization.

**Strengths:**

The reasoning tree formulation is intuitive and well motivated. The authors clearly articulate the limitations of path-based metrics and provide convincing evidence supporting the structural view of reasoning. Empirically, the results demonstrate consistent improvements: both Qwen2.5-Math-7B and Qwen2.5-7B models outperform all baselines across multiple mathematical reasoning datasets. The ablation studies and supporting analyses further validate the soundness of the approach.

**Weaknesses:**

1. Since reasoning trees can grow exponentially with depth, the authors approximate them using a fixed structure defined by (k,d,l). While this keeps the computation tractable, it may oversimplify the actual reasoning process. Although ablation studies explore nearby settings, all tested configurations remain within the same order of magnitude. How sensitive is performance to these parameters? Would larger or deeper trees provide further gains? Or would they introduce diminishing returns? Some intuition for choosing these parameters in advance would be valuable.
2. The paper does not fully discuss the computational cost of sampling k^d branches, even with KV-cache reuse. A quantitative analysis of runtime or memory cost would help assess scalability to larger models or non-mathematical reasoning tasks.
3. Writing and clarity: In Section 4, the meaning of the blue line in Figure 2 is unclear. Does it correspond to queries with more complex reasoning structures? Likewise, the terms potential samples and stagnant samples are not well explained in the caption. Clarifying these details would improve readability.
4. The R-score metric is intuitive but relies on several hyperparameters \eta,w_{\min},w_{\max},k,d,l, which may need careful tuning.

**Questions:**

1. In Figure 3(a), do the illustrated parameters correspond to k,d=3? It would be helpful to align the figure with the numerical example in Lines 256–257 for consistency.
2. What would happen if training started with lower R-score (i.e., more difficult) samples? Would the model still converge, and how would this affect generalization?
3. Since the evaluation temperature is set to 1.0, which introduces more randomness, how many evaluation runs were averaged? Are the reported improvements statistically significant?

---

> ### Author Response · Authors · 2025-11-24
> **Response to Reviewer qoyQ (1/3)**
>
> We sincerely thank the reviewer for the insightful feedback and for recognizing the motivation behind our reasoning tree formulation and the consistent empirical improvements of our method. We appreciate the opportunity to clarify the robustness of our hyperparameters, the computational trade-offs, and the details of our experimental setup. We have addressed your questions point-by-point below.
>
> ---
>
> ##  Weakness 1, 2 & 4: Hyperparameter Sensitivity and Scalability
>
> > - Since reasoning trees can grow exponentially with depth, the authors approximate them using a fixed structure defined by (k,d,l). While this keeps the computation tractable, it may oversimplify the actual reasoning process. Although ablation studies explore nearby settings, all tested configurations remain within the same order of magnitude. How sensitive is performance to these parameters? Would larger or deeper trees provide further gains? Or would they introduce diminishing returns? Some intuition for choosing these parameters in advance would be valuable.
> > - The paper does not fully discuss the computational cost of sampling k^d branches, even with KV-cache reuse. A quantitative analysis of runtime or memory cost would help assess scalability to larger models or non-mathematical reasoning tasks.
> > - The R-score metric is intuitive but relies on several hyperparameters \eta,w_{\min},w_{\max},k,d,l, which may need careful tuning.
>
> To address your concerns regarding the sensitivity and scalability of our tree construction parameters, we conducted an additional ablation study. Our results demonstrate the robustness of parameter $l$, while clarifying that parameters $k$ and $d$ represent a trade-off between performance and computational cost.
>
> We conducted parametric analysis experiments on k, d in **Section 7.3** of the original paper. Furthermore, during the rebuttal, we performed additional experiments on Qwen2.5-Math-7B to further verify these claims.
>
> As for $\omega$, the hyperparameter sensitivity analysis on the weight function hyperparameters has been shown in the **ablation study Table 4**.
>
> 1. **The Performance-Cost Trade-off** **($k$ and  $d$)** The parameters $k$ (branching factor) and $d$ (depth) define an approximation of the true, exponentially large reasoning tree.
>    1. **Performance:** As shown in the tables below, increasing $k$ or $d$ generally leads to better performance. A finer-grained tree captures structural difficulty more faithfully, yielding a more accurate r-score estimate; this, in turn, demonstrates that with sufficient compute, our method holds strong promise.
>    2. **Cost:** This improvement comes with a computational cost. While our KV-Cache reuse strategy significantly reduces redundancy, the sampling cost grows with $k^d$.
>    3. **Conclusion:** Our default setting ($k=4, d=4$) represents a "sweet spot" that balances high performance with manageable offline preprocessing costs. However, for scenarios with abundant compute, larger trees can unlock further gains.
>
> **(a) Varying Branching Factor ($k$)** (Fixed $d=4, l=200$)
>
> | Setting         | AIME24 | AIME25 | AMC23 | MATH500 | Minerva | Olympiad | Avg  |
> | --------------- | ------ | ------ | ----- | ------- | ------- | -------- | ---- |
> | Linear Mapping  |        |        |       |         |         |          |      |
> | $k=3$           | 32.5   | 15.2   | 74    | 81.8    | 36.3    | 41       | 46.8 |
> | $k=4$ (Default) | 34.2   | 15.6   | 72.4  | 81.2    | 36.4    | 42.5     | 47.1 |
> | $k=5$           | 35.1   | 18.1   | 77.4  | 81.8    | 37.8    | 44.6     | 49.1 |
> | Sigmoid Mapping |        |        |       |         |         |          |      |
> | $k=3$           | 32     | 15.2   | 74.2  | 81.6    | 38.6    | 43.1     | 47.5 |
> | $k=4$ (Default) | 34.2   | 16     | 71.1  | 81.8    | 42.3    | 44.4     | 48.3 |
> | $k=5$           | 36.4   | 17     | 75.5  | 81.2    | 40.6    | 43.4     | 49   |
>
> **(b) Varying** **Tree** **Depth ($d$)** (Fixed $k=4, l=200$)
>
> | Setting         | AIME24 | AIME25 | AMC23 | MATH500 | Minerva | Olympiad | Avg  |
> | --------------- | ------ | ------ | ----- | ------- | ------- | -------- | ---- |
> | Linear Mapping  |        |        |       |         |         |          |      |
> | $d=2$           | 31.1   | 15.3   | 74.2  | 81.8    | 38.2    | 42.2     | 47.1 |
> | $d=3$           | 31.4   | 14.6   | 72.7  | 82      | 39.7    | 43.3     | 47.3 |
> | $d=4$ (Default) | 34.2   | 15.6   | 72.4  | 81.2    | 36.4    | 42.5     | 47.1 |
> | Sigmoid Mapping |        |        |       |         |         |          |      |
> | $d=2$           | 31.9   | 14.8   | 74.5  | 81.6    | 37.2    | 42.4     | 47.1 |
> | $d=3$           | 33.2   | 16.4   | 73    | 80      | 41.9    | 41.6     | 47.7 |
> | $d=4$ (Default) | 34.2   | 16     | 71.1  | 81.8    | 42.3    | 44.4     | 48.3 |

---

> ### Author Response · Authors · 2025-11-24
> **Response to Reviewer qoyQ (2/3)**
>
> **(c) Computational Cost Analysis** We measured the offline construction time at 8 $\times$ H20 GPUs. For context, standard RL training takes approximately 9.34 hours per epoch (typically 5 epochs total).
>
> | Tree size $k^d$          | $3^3$  | $4^3$   | $3^4$   | $4^4$   |
> | ------------------------ | ------ | ------- | ------- | ------- |
> | Time cost (hours)        | 3.48   | 6.21    | 6.70    | 22.67   |
> | Additional time cost     | +7.45% | +13.30% | +14.35% | +48.54% |
> | Average performance gain | +3.2   | +3.0    | +3.2    | +4.0    |
>
> Additional time cost refers to the additional time overhead when constructing reasoning trees compared to the training process. This analysis confirms that while larger trees increase preprocessing time, the cost remains feasible, and researchers can select parameters based on their available resources. For example, under tight computational budgets, setting ($k=3$) and ($d=3$) incurs about $7%$ additional overhead relative to the baseline method, yet yields an average improvement of $3.2$, etc. In sum, we observe a clear trade-off: practitioners can flexibly choose the tree size according to available compute to realize performance gains.
>
> 1.  **Sensitivity to Token Interval ($l$)** The token interval $l$ determines the granularity of the tree.
>    1. **Robustness:** Our experiments (Table below) show that performance is stable across a wide range of $l$ ($200$ to $600$).
>    2. **Logic:** If $l$ is too large, the tree fails to branch before the answer concludes; if too small, it increases cost without adding significant structural information. We chose $l=200$ based on average solution lengths in math datasets.
>
> **Varying Token Interval ($l$)** (Fixed $k=4, d=4$)
>
> | Interval | Linear Mapping Avg | Sigmoid Mapping Avg |
> | -------- | ------------------ | ------------------- |
> | $l=200$  | 47.1               | 48.3                |
> | $l=400$  | 47.8               | 48.6                |
> | $l=600$  | 47.5               | 48.9                |
>
> These experimental content has been added to **Appendices C.3, C.4 and Section 7.4.1**.
>
> ## Weakness 3: Clarity of Figure 2
>
> > Writing and clarity: In Section 4, the meaning of the blue line in Figure 2 is unclear. Does it correspond to queries with more complex reasoning structures? Likewise, the terms potential samples and stagnant samples are not well explained in the caption. Clarifying these details would improve readability.
>
> Thank you for pointing this out. We have revised the caption and text in Section 4 to explicitly clarify these definitions:
>
> - Blue Line ("Stagnant Samples"): Represents queries with high initial accuracy but low r-scores (complex structures). As shown, their learning curve is flat, indicating that despite starting "well," they are difficult to improve further.
> - Red Line ("Potential Samples"): Represents queries with low initial accuracy but high r-scores (simple structures). Their learning curve is steep, indicating high "learnability"—a small amount of training yields significant gains.
>
> ## Question 1: Figure 3(a) Parameters
>
> > In Figure 3(a), do the illustrated parameters correspond to k,d=3? It would be helpful to align the figure with the numerical example in Lines 256–257 for consistency.
>
> Figure 3(a) is a schematic illustration intended to visualize the branching process conceptually. We intentionally used $k=3, d=3$ for visual clarity, as drawing the full $k=4, d=4$ tree would result in an overly cluttered and unreadable diagram. We have updated the figure caption to explicitly state that this is a simplified schematic and does not represent the exact experimental hyperparameters.

---

> ### Author Response · Authors · 2025-11-24
> **Response to Reviewer qoyQ (3/3)**
>
> ## Question 2: Reverse Curriculum
>
> > What would happen if training started with lower R-score (i.e., more difficult) samples? Would the model still converge, and how would this affect generalization?
>
> To investigate this, we conducted a "Reverse Schedule" experiment on Qwen2.5-Math-7B, where difficult (low r-score) samples were prioritized at the beginning of training.
>
> As shown in the table below, the Reverse Schedule results in consistent performance degradation across benchmarks compared to our standard Re-Schedule (e.g., a drop from **47.1% to 45.7%** in the Linear setting).
>
> | Schedule            | AIME24   | AIME25   | AMC23    | MATH500  | Minerva  | Olympiad | Avg      |
> | ------------------- | -------- | -------- | -------- | -------- | -------- | -------- | -------- |
> | **Linear Mapping**  |          |          |          |          |          |          |          |
> | Re-Schedule (Ours)  | 34.2 | 15.6 | 72.4 | 81.2 | 36.4     | 42.5 | 47.1 |
> | Reverse Schedule    | 31.9     | 14.0     | 67.6     | 81.0     | 37.8 | 41.8     | 45.7     |
> | **Sigmoid Mapping** |          |          |          |          |          |          |          |
> | Re-Schedule (Ours)  | 34.2 | 16.0 | 71.1     | 81.8 | 42.3 | 44.4 | 48.3 |
> | Reverse Schedule    | 30.2     | 15.4     | 67.1     | 80.6     | 34.9     | 40.2     | 44.7     |
>
> The failure of the Reverse Schedule highlights a critical characteristic of sampling-based RL algorithms like GRPO:
>
> 1. **Signal Starvation:** In the initial training phase, the model's policy is not yet optimized. If fed structurally complex (low r-score) queries immediately, the model fails to sample *any* correct solution paths. This leads to a lack of positive reward signals, effectively stalling the optimization process.
> 2. **Progressive Exploration:** By starting with high r-score (structurally simpler) queries, the model can easily find correct paths and obtain stable reward signals. This establishes a foundational policy, enabling the model to eventually explore and solve the more complex reasoning trees found in later stages.
>
> We have added these results and analysis to **Section 7.4.2**.
>
> ## Question 3: Evaluation Stability
>
> > Since the evaluation temperature is set to 1.0, which introduces more randomness, how many evaluation runs were averaged? Are the reported improvements statistically significant?
>
> Thank you for pointing this out. For all datasets, we report Avg@32 (averaged over random seeds). We have now updated **Section 7.1** to reflect these details.
>
> To further address your concerns regarding stability, we conducted experiments with different random seeds. The results, shown in the table below, demonstrate that our method exhibits low variance and consistent performance across runs. Compared to performance improvements, the impact of variance is minimal. This experimental content has been added to **Appendix C.2**.
>
> |                 | AIME24         | AIME25         | AMC23          | MATH500        | Minerva        | Olympiad       |
> | --------------- | -------------- | -------------- | -------------- | -------------- | -------------- | -------------- |
> | Avg. $\pm$ Var. | 35.2 $\pm$ 0.1 | 16.0 $\pm$ 0.0 | 72.3 $\pm$ 0.4 | 82.2 $\pm$ 0.0 | 42.3 $\pm$ 0.6 | 44.4 $\pm$ 0.6 |
>
> ---
>
> ## **Thanks again**
>
> We sincerely thank you again for your constructive and detailed feedback, which has helped us identify clear pathways to improve our paper. Should you have any further questions or require additional discussion, please don't hesitate to reach out. If we have adequately addressed your concerns, we would be grateful for your consideration in adjusting your evaluation score accordingly.

---

### Official Review · Reviewer_3KpD · 2025-11-01

**Soundness:** 2
**Presentation:** 2
**Contribution:** 2
**Rating:** 2
**Confidence:** 3

**Summary:**

This work proposes Reasoning Tree Schedule (Re-Schedule), a scheduling algorithm for reasoning tasks that constructs a curriculum from structurally simple to complex problems. The complexity is defined as r-score, a metric that measures the difficulty for learning based on the structure of the reasoning tree. Experiments on six math benchmarks show that Re-Schedule achieves better performance than existing methods, with an average improvement of up to 3.2%.

**Strengths:**

* This work introduces a novel metric, r-score, to measure the difficulty of reasoning tasks in terms of the structure of the reasoning tree. The proposed metric shifts the focus from only accuracy to structure-based difficulty assessment.
* With the proposed metric, the authors construct a curriculum from structurally simple to complex problems. This method is potentially applicable to other tasks and domains.
* The curriculum is used to train a reasoning model, which shows improved performance on six math benchmarks.
* The ablation studies show the importance of the proposed hyperparameters.

**Weaknesses:**

* Missing detailed analysis of the variation of the results. For example, since AIME 24/25 only has 30 problems each, the results may be not stable. The variation can be more than the 1.7% improvement on Qwen2.5-7B in Table 2.
* Adding to the point above, while prior works such as SimpleRL-Zoo (which was cited in Table 1 and 2) uses metrics such as Avg@32 to make the result more stable, the authors did not mention whether averaging is performs for Re-Schedule. If not performed, the results may be not stable.
* This work performs experiments with a fixed token interval (l=200). However, no ablations are performed to investigate the impact of the token interval, which determines how coarse the reasoning tree is.
* Comparisons with prior works are not performed in a fair way. For example, SimpleRL-Zoo makes use of GSM8k and MATH datasets only, while this work makes use of DAPO training set. Since the proposed method is a training schedule algorithm, the comparison should be performed in the same training set with the baseline (or with a reproduced result using the DAPO training set).
* Authors did not provide theoretical analysis of the proposed method. The proposed node-editing motivation is intuitive, but not formally justified or validated beyond empirical observation in Figure 2.

**Questions:**

* Are results obtained with averaging over multiple runs? If not, the authors are encouraged to provide additional experiments with averaging to make the results more stable. This affects the validity of the method and is an important factor of the score and recommendation.
* Does the optimal token interval depend on the model type and the difficulty of the question set (e.g., Qwen2.5-7B-Math vs Qwen2.5-7B, GSM8k vs DAPO training set)?

---

> ### Author Response · Authors · 2025-11-24
> **Response to Reviewer 3KpD (1/3)**
>
> We sincerely thank the reviewer for the constructive feedback and the recognition of our novel metric (r-score) and curriculum design. We appreciate the opportunity to clarify the stability of our results, the robustness of our hyperparameters, and the theoretical intuition behind our method. We address your concerns point-by-point below.
>
> ---
>
> ## **Weakness 1, 2 & Question 1: Result Stability and Averaging**
>
> > - Missing detailed analysis of the variation of the results. For example, since AIME 24/25 only has 30 problems each, the results may not be stable. The variation can be more than the 1.7% improvement on Qwen2.5-7B in Table 2.
> > - Adding to the point above, while prior works such as SimpleRL-Zoo (which was cited in Table 1 and 2) uses metrics such as Avg@32 to make the result more stable, the authors did not mention whether averaging is performs for Re-Schedule. If not performed, the results may be not stable.
> > - Are results obtained with averaging over multiple runs? If not, the authors are encouraged to provide additional experiments with averaging to make the results more stable. This affects the validity of the method and is an important factor of the score and recommendation.
>
> Thank you for pointing this out. For all datasets, we report Avg@32 (averaged over random seeds). We have now updated **Section 7.1** to reflect these details.
>
> To further address your concerns regarding stability, we conducted experiments with different random seeds. The results, shown in the table below, demonstrate that our method exhibits low variance and consistent performance across runs. Compared to performance improvements, the impact of variance is minimal. This experimental content has been added to **Appendix C.2**.
>
> |                 | AIME24         | AIME25         | AMC23          | MATH500        | Minerva        | Olympiad       |
> | ------------ | ---------- | ---------- | ---------- | ---------- | ----------- | --------- |
> | Avg. $\pm$ Var. | 35.2 $\pm$ 0.1 | 16.0 $\pm$ 0.0 | 72.3 $\pm$ 0.4 | 82.2 $\pm$ 0.0 | 42.3 $\pm$ 0.6 | 44.4 $\pm$ 0.6 |
>
> ## **Weakness 3 & Question 2: Token Interval Ablation**
>
> > - This work performs experiments with a fixed token interval (l=200). However, no ablations are performed to investigate the impact of the token interval, which determines how coarse the reasoning tree is.
> > - Does the optimal token interval depend on the model type and the difficulty of the question set (e.g., Qwen2.5-7B-Math vs Qwen2.5-7B, GSM8k vs DAPO training set)?
>
> **Regarding the sensitivity across different models:**
>
> To address your concerns, we conducted the ablation study on two different models: Qwen2.5-Math-7B and Qwen2.5-7B. We tested intervals ranging from $l=200$ to $l=1200$ using the Sigmoid mapping.
>
> Experimental Results (Average Accuracy):
>
> | Interval ($l$) | Qwen2.5-Math-7B (Avg) | Qwen2.5-7B (Avg) |
> | ---------- | ----------------- | ------------ |
> | $l=0$        | 44.3                  | 40.7             |
> | $l=200$      | 48.3                  | 44.5             |
> | $l=400$       | 48.6                  | 43.2             |
> | $l=600$       | 48.9                  | 43.1             |
> | $l=1200$      | 46.0                  | 41.1             |
>
> The results give two key observations:
>
> - **Robustness ($200 \le l \le 600$):** Our method demonstrates relatively stable performance within the range of $[200, 600]$. Over the $0-600$ interval, performance initially increases and then decreases. To minimize hyperparameter tuning, we found that setting a fixed $l = 200$ yields consistently good results across different models, continuing to outperform existing methods.
> - **Degradation at Coarse Granularity ($l=1200$):** When the interval becomes too large ($l=1200$), performance drops noticeably across both models. An excessively large **$l$** results in a reasoning tree that is too coarse, failing to capture critical decision points in the reasoning process.
>
> We have added these detailed ablation results to **Appendix C.3**.
>
> **Regarding the sensitivity** **across different datasets:**
>
> We acknowledge that analyzing the dataset sensitivity is complex due to the multifaceted factors involved. But since the average solution length naturally varies between datasets (e.g., GSM8k vs. OlympiadBench), the optimal $l$ would require adjustment to ensure the tree covers the full reasoning process without truncation.
>
> However, our results demonstrate consistent performance within a single dataset when $l$ remains in a reasonable range. As shown in the model ablation study, once a reasonable range is identified for a dataset (based on its average solution length), varying $l$ significantly (e.g., from 200 to 600) has minimal impact on performance. This implies that precise hyperparameter tuning is not necessary when dataset is given. For any given dataset, selecting a single, approximate $l$ is sufficient to achieve state-of-the-art results, further validating the practicality of our approach.

---

> ### Author Response · Authors · 2025-11-24
> **Response to Reviewer 3KpD (2/3)**
>
> ## **Weakness 4: Fair Comparison with Baselines**
>
> > Comparisons with prior works are not performed in a fair way. For example, SimpleRL-Zoo makes use of GSM8k and MATH datasets only, while this work makes use of DAPO training set. Since the proposed method is a training schedule algorithm, the comparison should be performed in the same training set with the baseline (or with a reproduced result using the DAPO training set).
>
> Yes, original SimpleRL-Zoo results were based on GSM8k and MATH, whereas our method utilized the DAPO training set. Thank you for pointing this out! We have updated the aligned results of the SimpleRL-Zoo in our paper. Notably, OPO, GRPO, LPPO, and Seed-GRPO datasets in our paper all use the DAPO-Math-17k training dataset, which strictly stick to the fair comparison.
>
> To ensure a strictly fair comparison, we reproduced SimpleRL-Zoo using the exact same training set (DAPO-Math-17k) as Re-Schedule.
>
> The comparison is presented below:
>
> Qwen2.5-Math-7B
>
> | Model                          | AIME24 | AIME25 | AMC23 | MATH500 | Minerva | Olympiad | Avg  |
> | ------------------------------ | ------ | ------ | ----- | ------- | ------- | -------- | ---- |
> | SimpleRL-Zoo (Original)        | 25.2   | 13.4   | 70.6  | 78.6    | 37.9    | 38.4     | 44.0 |
> | SimpleRL-Zoo (Trained on DAPO) | 30.8   | 14.2   | 65.4  | 79.2    | 37.1    | 40.8     | 44.6 |
> | Re-Schedule (linear)           | 34.2   | 15.6   | 72.4  | 81.2    | 36.4    | 42.5     | 47.1 |
> | Re-Schedule (sigmoid)          | 35.2   | 16.0   | 72.3  | 82.2    | 42.3    | 44.4     | 48.5 |
>
> Qwen2.5-7B
>
> | Model                                  | AIME24 | AIME25 | AMC23 | MATH500 | Minerva | Olympiad | Avg  |
> | -------------------------------------- | ------ | ------ | ----- | ------- | ------- | -------- | ---- |
> | SimpleRL-Zoo(Original)                 | 15.7   | 8.8    | 65.7  | 76.0    | 33.8    | 38.7     | 39.8 |
> | SimpleRL-Zoo(Trained on DAPO-Math-17k) | 17.0   | 9.6    | 64.7  | 76.6    | 31.6    | 40.3     | 40.0 |
> | Re-Schedule (linear)                   | 18.4   | 12.2   | 68.6  | 80.4    | 41.2    | 42.1     | 43.8 |
> | Re-Schedule (sigmoid)                  | 18.2   | 14.0   | 69.2  | 81.0    | 41.5    | 43.3     | 44.5 |
>
> Even when trained on the same data, Re-Schedule consistently outperforms SimpleRL-Zoo. In our paper, we have updated the SimpleRL-Zoo results in Tables 1 and 2.

---

> ### Author Response · Authors · 2025-11-24
> **Response to Reviewer 3KpD (3/3)**
>
> ## **Weakness 5: Theoretical Justification**
>
> > Authors did not provide theoretical analysis of the proposed method. The proposed node-editing motivation is intuitive, but not formally justified or validated beyond empirical observation in Figure 2.
>
> While a rigorous mathematical proof for Re-Schedule's effectiveness is challenging, we provide a formal qualitative framework supported by our empirical findings to justify the "node-editing" motivation.
>
> **1. Theoretical Framework: RLVR as** **Tree** **Optimization**
>
> We conceptualize RLVR training as an optimization problem where the goal is to maximize the expected accuracy gain over a batch of queries $B_t$. An ideal scheduler prioritizes samples with the highest expected gradient contribution:
>
> $$\max_{B_t} \sum_{q \in B_t} \mathbb{E}[\Delta \text{ACC}(q, \pi_t)] $$
>
> **2. Core Hypothesis: r-score as a Proxy for Learning Gain**
>
> We posit that the r-score $R(q$ serves as an effective proxy for this expected gain.
>
> - **High r-score:** Indicates that errors are "concentrated" in a few critical nodes. Correcting these specific nodes (efficient credit assignment) resolves large sub-trees of errors, yielding high learning gain.
> - **Low r-score:** Indicates "diffuse" errors scattered across the tree. The gradient signal is noisy or diluted, making it difficult for the policy to identify and fix the root cause, resulting in low learning efficiency.
>
>
> Therefore, Re-Schedule acts as a greedy strategy, prioritizing high r-score samples to maximize $\sum \mathbb{E}[\Delta \text{ACC}]$ in the early stages.
>
> **3. Empirical** **Validation**
>
> This framework is supported by our experiments:
>
> - **Figure 2** validates that high r-score samples ("Potential Samples") indeed exhibit steeper learning curves (higher $\Delta \text{ACC}$) compared to low r-score samples, even if their initial accuracy is lower.
> - **Section 6.1 (MCN Analysis)** shows that the Minimum Corrective Nodes (MCN) metric decreases during training. This confirms that RL effectively optimizes the tree structure. Since the goal is to minimize correction cost, starting with samples that inherently require fewer corrections (high r-score) is the most efficient path.
>
> ---
>
> ## **Thanks again**
>
> We sincerely thank you again for your constructive and detailed feedback, which has helped us identify clear pathways to improve our paper. Should you have any further questions or require additional discussion, please don't hesitate to reach out. If we have adequately addressed your concerns, we would be grateful for your consideration in adjusting your evaluation score accordingly.

---

### Author Response · Authors · 2025-11-27
**Follow-up on Rebuttal Submission**

Dear Reviewers,

We hope this message finds you well. We are writing to kindly follow up on our rebuttal and would be happy to provide any additional clarification or information if needed. Thank you very much for your time and consideration.

Sincerely,

The Authors

---

### Author Response · Authors · 2025-12-02
**A Summary of Review, Rebuttal, and Discussion**

Dear Area Chair and Reviewers,

Thank you for the thoughtful and constructive review process. The feedback has helped us improve the manuscript. During the discussion, several key strengths were consistently recognized:

- **Novel and Principled Metric (r-score):**
  - Reviewers praised the shift from path-based accuracy to structural complexity. The **Reasoning Score (r-score)** was highlighted as a "more principled approach to capturing the learning dynamics" (`Reviewer Hh8P`) and a "valuable reframing" that addresses the limitations of accuracy-based metrics (`Reviewer 3KpD`, `Reviewer qoyQ`).
  - The "node-editing" analogy for RLVR was described as "highly intuitive" and well-motivated (`Reviewer Hh8P`, `Reviewer qoyQ`).
- **Strong Empirical Performance:**
  - The method’s ability to achieve state-of-the-art results across six math benchmarks was recognized as "compelling" (`Reviewer Hh8P`) and "consistent" (`Reviewer qoyQ`).
  - The curriculum design (Re-Schedule) was noted for effectively balancing exploration and exploitation (`Reviewer qoyQ`, `Reviewer eZz4`).

During the rebuttal, we went beyond clarification and substantially strengthened the paper with extensive new experiments:

- **Generalization to New Domains and Models:**
  - We extended our evaluation to **Code Generation** (training on ArcherCodeR, evaluating on LiveCodeBench), where Re-Schedule outperformed baselines, demonstrating that r-score generalizes beyond math. (for `Reviewers Hh8P, eZz4`)
  - We validated the method on a different model architecture (Qwen3-4B-Base), confirming the approach is model-agnostic. (for `Reviewer eZz4`)
- **Robustness, Stability, and Fair Comparison:**
  - We have addressed concerns regarding baseline fairness by reproducing SimpleRL-Zoo on the exact same training set (DAPO-Math-17k). Re-Schedule maintained its superiority. (for `Reviewer 3KpD`)
  - The original manuscript omitted the Avg@32 setting, leading reviewers to mistakenly assume single-run experiments. We have now explicitly clarified this in the revision. (for `Reviewers 3KpD, qoyQ`)
- **In-depth Ablation and Cost Analysis:**
  - We conducted comprehensive ablation studies on tree parameters ($k, d, l$) and provided a detailed **computational cost analysis**, showing that the offline construction cost is manageable and offers a clear performance-compute trade-off. (for `Reviewers qoyQ, Hh8P, eZz4`)
  - We empirically validated the "Easy-to-Hard" design via a Reverse Schedule experiment (which failed due to signal starvation) and confirmed that a Static r-score is sufficient compared to a computationally expensive Dynamic update. (for `Reviewers qoyQ, eZz4`)

During the discussion, Reviewer Hh8P explicitly noted that the revision "**resolved my concerns**". Since these specific issues, like generalization and robostness, were also the primary concerns raised by other reviewers, we believe our rebuttal has effectively addressed the major concerns of the reviewers.

We have incorporated all new results into **the revised manuscript** and are committed to open-sourcing our code and data to facilitate reproducibility.

Thank you again for your consideration.

Best regards,

Authors

---

### Meta-Review · Area_Chair_PtgS · 2026-01-07

**Summary:**

Overall, the reviewers’ concerns are largely addressed by the authors’ rebuttal. One remaining limitation is that, under the default setting, the proposed method incurs substantially higher computational cost compared to standard reinforcement learning approaches.

Despite this limitation, the paper’s strengths outweigh its weaknesses, and I recommend acceptance.

As an additional observation, the proposed tree structure bears resemblance to Monte Carlo Tree Search and could potentially be extended to incorporate an explicit confidence term. This connection is not discussed in the paper but does not materially affect the acceptance recommendation.

**Reviewer Concerns:**

The reviewers’ concerns have been adequately addressed, and no outstanding issues remain.

**Reviewer 3KpD**

- Weakness 1, 2 / Question 1: Result Stability and Averaging.

Addressed.

- Weakness 3 / Question 2: Token interval ablation.

Mostly addressed. The authors gave rationale on the difficulty of conducting sensitivity
analysis across different datasets.

- Weakness 4: Fair comparison with baselines.

Addressed.

- Weakness 5: Theoretical justification.

Somewhat addressed by intuitive rationale.

**Reviewer qoyQ**

- Weakness 1, 2, 4: Hyperparameter sensitivity and scalability.

Addressed, though the results do show that computational cost is >2x higher under the
default setting, which is a drawback of the proposed algorithm).

- Weakness 3 / Question 1: Presentation and clarity.

Addressed.

- Question 2: Reverse curriculum.

Addressed.

- Question 3: Evaluation stability.

Addressed.

**Reviewer Hh8P**

All comments are well-addressed.

**Reviewer eZz4**

All comments are well-addressed.

**Reviewer Scores:**

Reviewers Hh8P and eZz4 will likely keep the score of 6.

Reviewer 3KpD will probably increase the score from 2 to 4, but very unlikely from 2
to 6.

It is possible for Reviewer qoyQ to increase the score from 4 to 6.

---

### Decision · Program_Chairs · 2026-01-26

Accept (Poster)